# Feature-based learning improves adaptability without compromising precision

Shiva Farashahi[1], Katherine Rowe [1], Zohra Aslami[1], Daeyeol Lee [2,3,4,5] & Alireza Soltani [1]

Learning from reward feedback is essential for survival but can become extremely challenging with myriad choice options. Here, we propose that learning reward values of individual features can provide a heuristic for estimating reward values of choice options in dynamic, multi-dimensional environments. We hypothesize that this feature-based learning occurs not just because it can reduce dimensionality, but more importantly because it can increase adaptability without compromising precision of learning. We experimentally test this hypothesis and find that in dynamic environments, human subjects adopt feature-based learning even when this approach does not reduce dimensionality. Even in static, low-dimensional environments, subjects initially adopt feature-based learning and gradually switch to learning reward values of individual options, depending on how accurately objects' values can be predicted by combining feature values. Our computational models reproduce these results and highlight the importance of neurons coding feature values for parallel learning of values for features and objects.

[1] Department of Psychological and Brain Sciences, Dartmouth College, Hanover, NH 03755, USA. [2] Department of Neuroscience, Yale School of Medicine, New Haven, CT 06510, USA. [3] Kavli Institute for Neuroscience, Yale School of Medicine, New Haven, CT 06510, USA. [4] Department of Psychiatry, Yale School of Medicine, New Haven, CT 06511, USA. [5] Department of Psychology, Yale University, New Haven, CT 06520, USA. Shiva Farashahi and Katherine Rowe contributed equally to this work. Correspondence and requests for materials should be addressed to A.S. (email: soltani@dartmouth.edu)

Human behavior is marked by a sophisticated ability to attribute reward outcomes to appropriate choices and events with surprising nuance. Learning from reward feedback is essential for survival but can be extremely challenging in natural settings because choices have many features (e.g., color, shape, and texture), each of which can take different values, resulting in a large number of options for which reward values must be learned. This is referred to as the "curse of dimensionality," because the standard reinforcement learning (RL) models used to simulate human learning do not scale up with the increasing dimensionality[1–4].

An increase in dimensionality creates two main difficulties for humans and the standard RL models that attempt to directly learn the value of individual options. First, such learning is too slow because a large amount of reward feedback is needed for an accurate estimate of all reward values, resulting in imprecise estimates of reward value if reward contingencies quickly change over time. For example, a child naturally learns the tastes of various fruits she consumes throughout her life (e.g., green crispy apples, red crispy apples, yellow mushy bananas, etc.), but it would take a long time to acquire preferences for all different types of fruits. Second, the value of unexperienced options cannot be known; for example, how should the child approach a green, mushy avocado never encountered before?

A few approaches are proposed for how we overcome the curse of dimensionality. One approach is to construct a simplified representation of the stimuli and therefore, to learn only a small subset of features and ignore others[5, 6]. However, there are behavioral and neural data suggesting that in order to make decisions in multi-dimensional tasks, humans process all features of each option simultaneously, rather than focus on any single feature[7]. Moreover, ignoring certain features could be detrimental in dynamic environments where previously non-informative features can suddenly become informative. Another approach is to combine multiple environmental states or actions, thereby reducing the number of states or actions to be learned[8, 9]. Finally, one could infer the structure of the task and create rules to estimate reward values of options based on their features, which requires a much smaller set of values to be learned[10–13].

A simple form of this rule-based approach is feature-based learning, in which the reward values of all features are learned in parallel, and then combined according to a specific rule for estimating the reward values for individual options. For example, a child could evaluate fruits based on their color and texture and learn about these features when she consumes them. This heuristic feature-based learning is only beneficial if a generalizable set of rules exist to construct the reward value of all options accurately by combining the reward values of their features. Unfortunately, this is often not the case; for example, not all green fruits are tasty. So, could the benefits of feature-based learning overcome a lack of generalizable rules and still make this learning approach a viable heuristic? Currently, there is no single, unified framework for describing how such properties of the environment influence learning strategy (e.g., feature based vs. object based).

An important aspect of feature-based learning is that reward values of all features of the selected option can be updated based on a single reward feedback, as opposed to updating only the value of the selected option in object-based learning. This makes feature-based learning faster and more adaptable, without being noisier, than object-based learning. This is important because simply increasing the learning rates in object-based learning can improve adaptability but also adds noise in the estimation of reward values, which we refer to as the adaptability-precision tradeoff[14, 15]. Therefore, the main advantage of heuristic feature-based learning might be to mitigate the adaptability-

precision tradeoff. To test this hypothesis, we propose a general framework for understanding the advantages of feature-based learning and design a series of experiments to characterize how multiple factors encourage the adoption of feature-based vs. object-based learning. These factors include: dimensionality, or the number of options or features to be learned; and generalizability, or how well reward values of options can be estimated from the values of their features. Moreover, we construct and test two alternative network models to elucidate neural mechanisms consistent with our experimental observations.

We found that in dynamic environments, humans adopted feature-based learning even when this approach did not reduce dimensionality, namely, when the same numbers of options and features have to be learned. Even in static, low-dimensional environments where dimensionality reduction due to feature-based learning was small, subjects initially adopted feature-based learning and only gradually switched to learning individual option/object values. The degree of switching to object-based learning, however, was much smaller with higher dimensionality, or when objects' values could be more accurately predicted by combining the reward values of their features (i.e., higher generalizability). Overall, these results confirm our hypothesis and suggest feature-based learning as a powerful heuristic for learning in dynamic, multi-dimensional environments. Finally, we found that our experimental results can be better captured by a model that has separate systems for estimating reward values of objects and features, and uses the output of the system that carries a stronger singal to make a decision on a given trial and accordingly adjusts the weight of this system based on reward feedback.

## Results

**Feature-based learning mitigates adaptability-precision tradeoff.** To test our hypothesis that feature-based learning is mainly adopted to mitigate the adaptability-precision tradeoff, we first developed a general framework for learning in dynamic, multi-dimensional environments (see Methods for more details). If options/objects contain $m$ features, each of which can have $n$ instances, there would be $n^m$ possible objects in the environment. The decision maker's task is to learn the reward values of options/objects via reward feedback in order to maximize the total reward when choosing between two alternative options on each trial. To examine the advantages of object-based and feature-based approaches, we simulated this task using two different model learners. The object-based learner directly estimates the reward values of individual objects via reward feedback, whereas the feature-based learner estimates the reward values of all feature instances, such as red, blue, square, or triangle. The latter is achieved by updating the reward values associated with all features of the object for which reward feedback is given. The feature-based learner then combines the reward values of features to estimate the reward values of individual objects. To examine how the performance of the two learners depends on the reward statistics in the environment, we varied the relationship between the reward value of each object and the reward values of its features in order to generate multiple environments, each with a different level of generalizability. In a fully generalizable environment, the estimated reward probabilities based on features deviate from the actual reward probabilities by only a small degree (Supplementary Fig. 1), but more importantly, the rank order of estimated and actual reward probabilities, which determines preference between objects, is also similar.

Feature-based learning might be faster than object-based learning with the same learning rate because reward values of all features of the selected option can be updated after each reward feedback. In contrast, only the value of the selected option

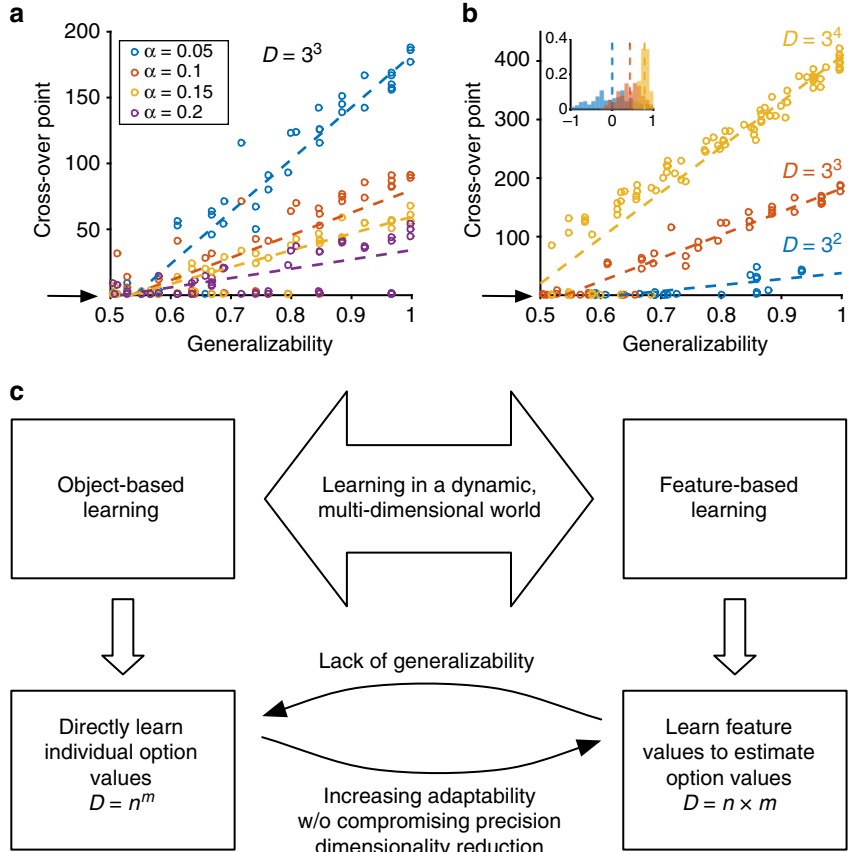

**Fig. 1** A framework for understanding model adoption during learning in dynamic, multi-dimensional environments. **a** Cross-over point is plotted as a function of the generalizability index of the environment for different values of the learning rate. The cross-over point increases with generalizability and decreases with the learning rate. The larger learning rate, however, comes at the cost of more noise in estimation (lower precision). The arrow shows zero cross-over point indicating that the object-based learning is always superior for certain environments. **b** Cross-over point is plotted as a function of generalizability separately for environments with different values of dimensionality (for $\alpha = 0.05$). The advantage of feature-based over object-based learning increases with larger dimensionality. The inset shows the distribution of the generalizability index in randomly generated environments for three different dimensionalities. **c** The object-based approach for learning multi-dimensional options/objects requires learning $n^m$ values, where there are $m$ possible features and $n$ instances per feature in the environment, whereas the feature-based approach requires learning only $n \times m$ values resulting in a dimensionality reduction equal to ($n^m - n \times m$). A feature-based approach, however, is beneficial if there are generalizable rules for estimating the reward values of options based on the combination of features' values. A lack of generalizability should encourage using the object-based approach. Finally, frequent changes in reward contingencies (dynamic environment) should increase the use of feature-based learning because it allows update of multiple features based on a single feedback and thus increases adaptability without compromising precision

is updated in the object-based learning model. Given a sufficient amount of time, the object-based learner can accurately estimate all option values, whereas the accuracy of the feature-based learner is limited by the generalizability of the environment. By comparing the time course of information acquired by the object-based and feature-based learners when reward values are fixed and using the same learning rate, we computed the time at which the object-based learner acquires more information than the feature-based learner (the 'cross-over point'; Methods section).

We found that for sufficiently large values of generalizability (>0.5), the feature-based learner acquires more information early on, but ultimately, the object-based learner reaches the same level of information as the feature-based learner and later surpasses it. Thus, object-based learning will be ultimately more useful in a stable environment. On the other hand, feature-based learning might be more beneficial in volatile environments where reward contingencies change often. Moreover, the cross-over point occurs later for smaller learning rates, indicating that slowing down learning to increase precision would favor feature-based learning (Fig. 1a). The advantage of feature-based over object-based learning increases with the dimensionality of the

environment, as the number of value updates per reward feedback increases with the number of features in each object (Fig. 1b). Finally, an environment with randomly assigned reward probabilities tends to be more generalizable as the dimensionality increases (Fig. 1b, inset). This property further increases the advantage of adopting feature-based learning in high-dimensional environments.

These simulations demonstrate how the adaptability-precision tradeoff might favor the adoption of feature-based over object-based learning in certain environments. Because only the value of the selected option is updated after each reward feedback, object-based learning in a volatile environment requires a higher learning rate, which comes at the cost of lower precision. Feature-based learning can mitigate this problem by speeding up the learning via more updates per feedback, instead of increasing the learning rate.

Our simple framework also provides clear predictions about how different factors such as dimensionality reduction, generalizability, and volatility might influence the adoption of feature-based learning. Frequent changes in reward contingencies and high dimensionality should force the decision maker to adopt

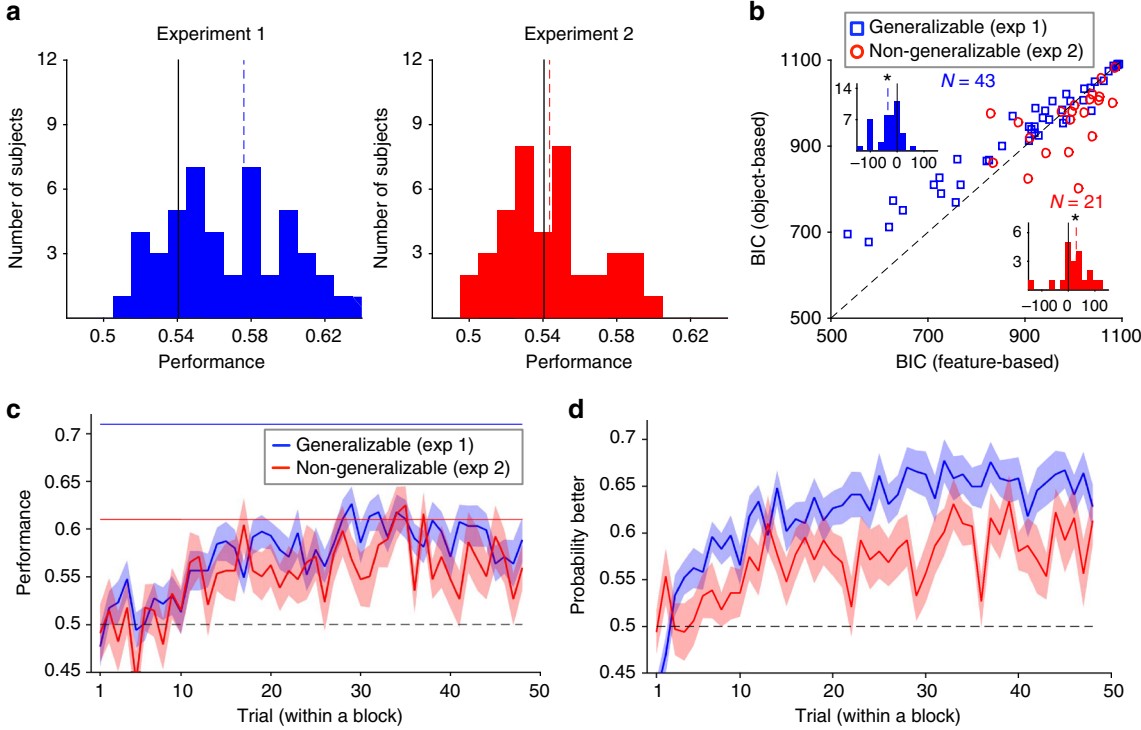

**Fig. 2** Dynamic reward schedules promote feature-based learning whereas a lack of generalizability promotes object-based learning. **a** Performance or the average reward harvested by subjects during Experiments 1 (generalizable environment) and 2 (non-generalizable environment). Dashed lines show the mean performance and solid lines show the threshold used for excluding subjects whose performance was not distinguishable from chance (0.5). **b** Plotted is the Bayesian information criterion (BIC) based on the best feature-based or object-based models, separately for each environment. The insets show histograms of the difference in BIC from the two models for the generalizable (blue) and non-generalizable (red) environments. The dashed lines show the medians and the stars indicate significant difference from zero (two-sided rank-sum, $P < 0.05$). Subjects were more likely to adopt a feature-based approach in the generalizable environment and an object-based approach in the non-generalizable environment. **c, d** Time course of learning during each block of trials in Experiments 1 and 2. Plotted are the average harvested reward (**c**) and probability of selecting the better option (**d**) in a given trial within a block across all subjects (the shaded areas indicate s.e.m.). The dashed line shows chance performance. The solid blue and red lines show the maximum performance based on the feature-based approach in the generalizable and non-generalizable environments, respectively, assuming that the decision maker selects the more rewarding option based on this approach on every trial. The maximum performance for the object-based approach was similar in the two environments, and equal to that of the feature-based approach in the generalizable environment

feature-based learning (Fig. 1c). On the other hand, lack of generalizability of the reward values of features to all object values should encourage adopting more accurate object-based learning. But immediately after changes in reward values, feature-based learning should still be favored since it acquires reward information more quickly. We tested these predictions in four experiments.

**Feature-based learning in dynamic environments**. To test our hypothesis and explore different factors influencing model adoption in dynamic multi-dimensional environments, we designed four experiments in which human subjects learned the reward values of different objects through reward feedback. In all experiments, subjects chose between a pair of dissimilar objects associated with different reward probabilities, but the relationship between the reward probabilities of objects and those of their features (color, shape, etc.) was varied.

In Experiment 1, the pair of objects in each trial consisted of colored shapes with associated reward probabilities that unpredictably changed over time (between blocks of 48 trials; Supplementary Fig. 2). Importantly, the feature-based and object-based approaches required learning the same number of reward values: four objects (red square, red triangle, blue square, and blue triangle) and four feature instances (red, blue, square, and triangle). Therefore, adopting feature-based learning did not reduce dimensionality in Experiment 1. Moreover, reward

probabilities assigned to different objects, which we collectively refer to as the reward schedule, could be reliably estimated by combining the reward values of their features if the environment was generalizable. By examining choice behavior during Experiment 1, we aimed to study specifically how adaptability required in a dynamic environment influences the adoption of a model used for learning and decision making (Fig. 1c). Experiment 2 was similar to Experiment 1, except that reward probabilities assigned to different objects were not generalizable and could not be estimated accurately by combining the reward values of their features. Therefore, choice behavior in Experiment 2 could reveal how the adaptability required in a dynamic environment and a lack of generalizability both influence model adoption (Fig. 1c). Finally, in Experiments 3 and 4, we increased the dimensionality of the environment to examine the effect of a small and moderate dimensionality reduction by feature-based learning. Reward probabilities, however, were fixed throughout both of these experiments and reward values assigned to features were not fully generalizable to objects. This design allowed us to study the influence of dimensionality reduction and lack of generalizability on model adoption (Fig. 1c).

Overall, most subjects (64 out of 92) performed above the statistical chance level in both Experiments 1 and 2, indicating that they learned the values of options as they changed over time (Fig. 2a). To examine the time course of learning, we computed the average probability of reward as well as the probability of

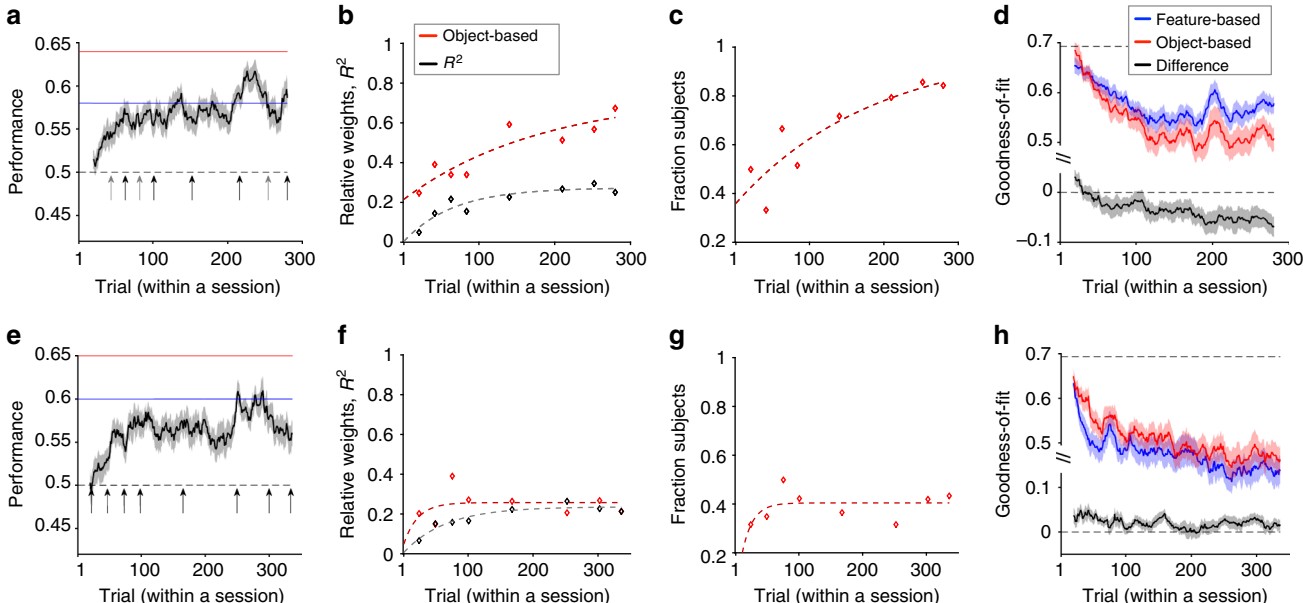

**Fig. 3** Transition from feature-based to object-based learning in static, non-generalizable environments. **a** The time course of performance during Experiment 3. The running average over time is computed using a moving box with the length of 20 trials. Shaded areas indicate s.e.m., and the dashed line shows chance performance. The red and blue solid lines show the maximum performance using the feature-based and object-based approaches, respectively, assuming that the decision maker selects the more rewarding option based on a given approach in every trial. Arrows mark the locations of estimation blocks throughout a session. For some subjects, there were only five estimation blocks indicated by black arrows. **b** The time course of model adoption measured by fitting subjects' estimates of reward probabilities. Plotted is the relative weight of object-based to the sum of the object-based and feature-based approaches, and explained variance in estimates ($R^2$) over time. Dotted lines show the fit of data based on an exponential function. **c** Plotted is the fraction of subjects who showed a stronger correlation between their reward estimates and actual reward probabilities than the probabilities estimated using the reward values of features. The dotted line shows the fit of data based on an exponential function. **d** Transition from feature-based to object-based learning revealed by the average goodness-of-fit over time. Plotted are the average negative log likelihood based on the best feature-based model, best object-based RL model, and the difference between object-based and feature-based models during Experiment 3. Shaded areas indicate s.e.m., and the dashed line shows the measure for chance prediction. **e–h** The same as in **a–d**, but during Experiment 4.

selecting the more rewarding option in a given trial during each block of trials (when probabilities were fixed). The latter quantity measured how well the subjects discriminated between the four options based on their associated reward probabilities. This analysis revealed that on average, it took approximately 15 trials for a subject to reach maximum performance (Fig. 2c) or discrimination (Fig. 2d) after a reversal. Examining choice behavior and performance in different super-blocks of the experiments did not reveal any significant change in achieving this performance over the course of the experiment (Supplementary Fig. 3a–d) but showed an overall decrease in learning in the last super-block of Experiment 2 (Supplementary Fig. 3a–d). Altogether, these results indicate that subjects did not use information from early reward schedules to predict future changes in reward schedules, a challenging task given that reward values for all four options changed between blocks.

To identify the learning model adopted by each subject, we fit the experimental data in each environment using six RL models that relied on either an object-based or a feature-based approach (Methods). To ensure that this fitting procedure can actually detect a specific learning approach adopted by individual subjects, we generated choice data using each of the six models over a wide range of model parameters and fit the resulting data with all the models (Methods section). This analysis demonstrates that our fitting method is able to distinguish between the alternative models used for generating the data (Supplementary Fig. 4) and to estimate underlying parameters accurately (Supplementary Fig. 5).

By fitting subjects' choice data, we found that the coupled feature-based RL (coupled indicates that values of both chosen

and unchosen options are updated in each trial) and feature-based RL with decay provided the best overall fits for the data in the generalizable environment (Experiment 1; Supplementary Table 1). More importantly, all feature-based models provided a better overall fit than their object-based counterparts. We also compared the goodness-of-fit based on the best feature-based and object-based models for each individual and found that feature-based models provided significantly better fits in the generalizable environment (BIC (best feature-based)−BIC (best object-based) (mean±s.d.) = −34.61 ± 47.55; two-sided sign-rank test, $P = 3.2 \times 10^{-5}$, $N = 43$, $d = 0.73$; Fig. 2b). By contrast, in the non-generalizable environment (Experiment 2), the object-based models provided significantly better fits than feature-based models (BIC (best feature-based)−BIC (best object-based) (mean±s.d.) = 29.77 ± 71.07, two-sided sign-rank test, $P = 0.030$, $N = 21$, $d = 0.42$). We found consistent results when we considered each of the four super-blocks separately, indicating that the observed pattern of model adoption was not due to the use of different strategies early and late in the experiments (Supplementary Fig. 6a–d). Finally, we analyzed choice behavior of the excluded subjects but did not find any evidence that those subjects adopted a unique strategy. These subjects did not favor feature-based or object-based learning, nor did they change their approach over the course of experiments. Instead, they likely did not engage in the experiment due to task difficulty, especially during Experiment 2 (Supplementary Figs. 6e–h and 7a–c).

Together, these results illustrate that subjects tended to adopt feature-based learning in the generalizable environment and object-based learning in the non-generalizable environment. Therefore, a dynamic reward schedule encouraged subjects to

use feature-based learning, which improves adaptability without compromising precision, whereas a lack of generalizability led them to switch (as a group) to slower but more accurate object-based learning.

**Shift from feature-based to object-based learning**. Our framework predicts that feature-based learning should be adopted initially until the acquired information derived from the object-based approach becomes comparable to information derived from the feature-based approach. To test this prediction, we designed two additional experiments (Experiments 3 and 4) in which human subjects learned the values of a larger set of objects in a static, non-generalizable environment (Methods section and Supplementary Fig. 8). The purpose of the static environment was to isolate the influence of generalizability and dimensionality reduction on model adoption in the absence of changes in reward schedules studied in Experiments 1 and 2. Moreover, in order to assess the temporal dynamics of model adoption more directly, we asked subjects to provide their estimates of reward probabilities for individual objects during five or eight estimation blocks throughout the experiment. The reward assignment was such that one of the two features was partially informative about the reward value, while the other feature did not provide any information by itself, resulting in non-generalizability of the environments. More specifically, the average reward values for instances of the non-informative feature were identical (Experiment 3) or very similar (Experiment 4) but the same average values for the informative feature were distinct (compare the average of values in individual columns or rows in Supplementary Fig. 8a).

Overall, the subjects were able to learn the task in Experiment 3, and the average performance across all subjects monotonically increased over time and plateaued at about 100 trials (Fig. 3a). Examination of the estimated reward probabilities for individual objects also showed an improvement over time, but more importantly, suggested a transition from a feature-based to an object-based approach as the experiment progressed. We utilized general linear regression and correlation to identify the model adopted by the subjects over the course of the experiment from their reward probability estimates (Methods section). The fit of subjects' estimates revealed that the weight of the object-based approach relative to the sum weights of the object-based and feature-based approaches was much smaller than 0.5 during the first estimation block but gradually increased over time (relative weight = 0.32, 95% CI (0.27 0.36) and 0.62, 95% CI (0.57 0.66) for the first two and last two estimates, respectively; Fig. 3b). In addition, the percentage of variance in estimates explained by object-based and feature-based approaches ($R^2$) gradually increased over the course of the experiment. Similarly, correlation analysis revealed that during early estimation blocks, the estimates of only a small fraction of subjects were more correlated with actual reward probabilities than reward probabilities estimated based on features, but this fraction increased over time (comparison of fractions in first two estimates vs. last two estimates: $\chi^2$ (1) = 17.14, $P = 3.5 \times 10^{-5}$, $N = 38$) (Fig. 3c). The results of these two analyses illustrated that in Experiment 3, subjects initially adopted feature-based learning and gradually switched to object-based learning.

This result indicates a transition from feature-based to object-based learning. In Experiment 4, however, feature-based RL with decay provided the best overall fit (Supplementary Table 1) and the fit of this model was better than the corresponding object-based learning model throughout the experiment (Fig. 3h). Overall, the results based on fitting choice behavior were consistent with the results based on subjects' reward estimates.

Finally, we performed similar analyses for choice behavior and estimation of the excluded subjects but did not find any evidence that those subjects adopted a strategy qualitatively different from the one used by the remaining subjects (Supplementary Fig. 7d–i). We increased dimensionality of the environment in Experiment 4 in relation to Experiment 3 to further examine the influence of dimensionality reduction on model adoption. The performance again plateaued at about 100 trials (Fig. 3e). Moreover, the fit of subjects' estimates revealed that the relative weight of the object-based approach only slightly increased over time and plateaued at a small value (relative weight = 0.17, 95% CI (0.13 0.21) and 0.24, 95% CI (0.21 0.27) for the first two and last two estimates, respectively; Fig. 3f). Correlation analysis revealed a very similar pattern when the fraction of subjects using object-based learning did not significantly increase over time (comparison of fractions in first two estimates vs. last two estimates: $\chi^2$ (1) = 2.27, $P = 0.13$, $N = 50$); Fig. 3g). All of these results suggest stronger feature-based learning compared to object-based learning when dimensionality or generalizability increased, because both these quantities increased in Experiment 4 relative to Experiment 3 (for Experiments 3 and 4, $D = 9$ and 16 and generalizability = 0.57 and 0.76, respectively).

We also fit the data from Experiments 3 and 4 using various RL models in order to identify the model adopted by the subjects. In Experiment 3, object-based RL with decay provided the best overall fit (Supplementary Table 1). Importantly, this model provided a better fit than its corresponding feature-based RL. Examination of the goodness-of-fit over time illustrated that the object-based learning model provided a better fit, particularly later in the experiment (Fig. 3d). The difference between the quality of the fit of the object-based and feature-based models in early (1–100) and late (100–280) trials ($\langle -LL_{object-based} + LL_{feature-based} \rangle_{early}$ $-\langle -LL_{object-based} + LL_{feature-based} \rangle_{late}$ (mean±s.d.) = 0.023 ± 0.050) was significantly different from zero (two-sided sign-rank test; $P = 0.029$, $d = 0.46$, $N = 27$). We note that the boundary for early vs. late trials (at 100) was selected based on the time course of performance (Fig. 3a) but that the reported difference was significantly larger than zero ($P < 0.05$) for any boundary values between 60 and 130 as well.

Together, we found that during both Experiments 3 and 4, subjects first adopted feature-based learning. In Experiment 3, they subsequently transitioned to object-based learning. Such a transition was not evident in Experiment 4 due to higher dimensionality and larger generalizability in Experiment 4, both of which would encourage feature-based learning.

**Testing the behavioral predictions of feature-based learning**. Feature-based learning assumes that reward feedback on a given object is attributed to all features of that object and thus, predicts that the reward value of all objects that share a feature with the object for which reward feedback was received should be updated. To test this prediction, we computed the feature-based 'differential response' to reward feedback that measures the differential change in the value of features of the object selected on the previous trial for when it was rewarded vs. when it was not rewarded (Methods section). We predicted that this measure would be positive for subjects who adopted feature-based learning. For comparison, we also calculated the object-based differential response equal to the difference between the probability of selecting an object that was selected and rewarded on the previous trial and the same probability when the previous trial was not rewarded. This measure is equivalent to the difference between win–stay and lose-stay strategy and should be positive for all subjects independently of their adopted model. We used goodness-of-fit to determine whether a subject adopted feature-based or object-based learning in a given experiment.

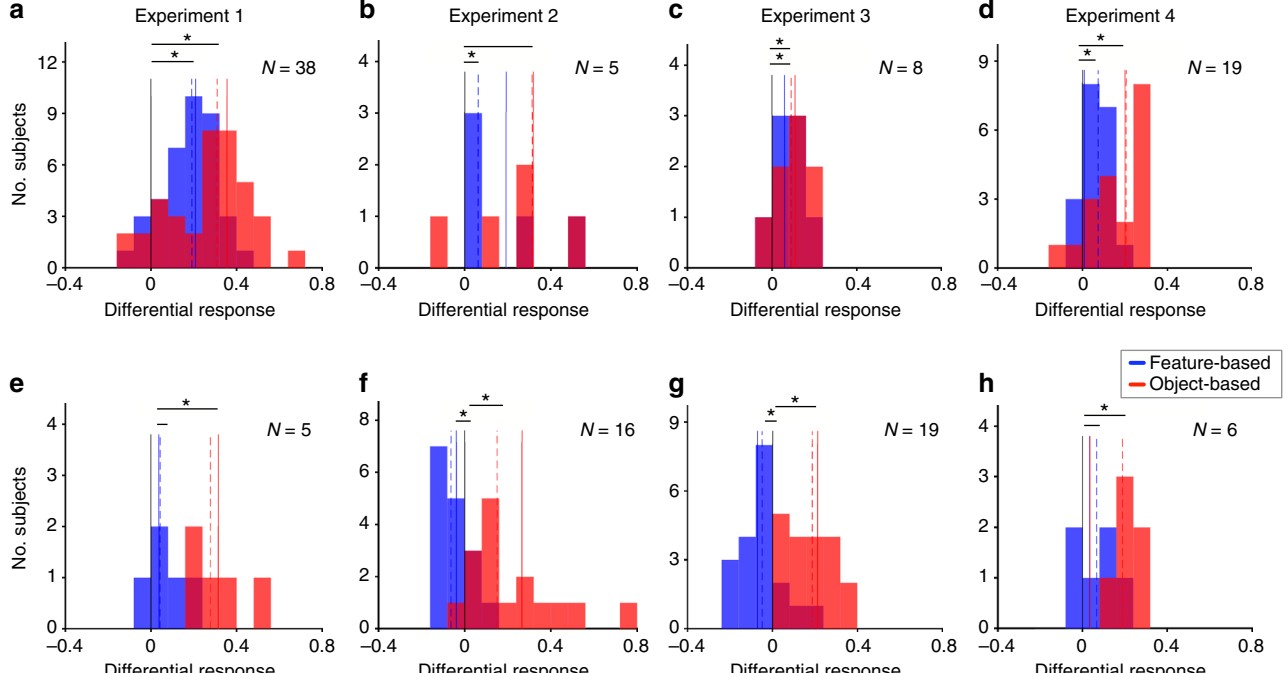

**Fig. 4** Subjects who adopted feature-based learning updated their preference even for other objects that contained a feature of the object selected on the previous trial based on the reward outcome. **a–d** Plotted are the feature-based (blue) and object-based (red) differential responses for subjects who adopted feature-based learning in a given experiment. The dashed lines show the median values across subjects and a star indicates significant difference from zero (one-sided sign-rank test, $P < 0.05$). The solid lines show the average simulated differential response using the estimated parameters based on the fit of each subject's data. **e–h** The same as in **a–d** but for subjects who adopted object-based learning in each experiment

The feature-based differential response was overall positive for subjects who adopted feature-based learning in all four experiments, but this effect was more difficult to observe in Experiments 2 and 3 due to the small sample size, since most subjects adopted object-based learning in these experiments (one-sided sign-rank test; Experiment 1: $P = 1.3 \times 10^{-7}$, $d = 1.48$, $N = 38$; Experiment 2: $P = 0.031$, $d = 0.97$, $N = 5$; Experiment 3: $P = 0.012$, $d = 1.07$, $N = 8$; Experiment 4: $P = 0.0065$, $d = 1.004$, $N = 19$; Fig. 4a–d). In contrast, feature-based differential response was overall negative or undistinguishable from 0 for subjects who adopted object-based learning (one-sided sign-rank test, Experiment 1: $P = 0.16$, $d = 0.70$, $N = 5$; Experiment 2: $P = 0.01$, $d = 0.62$, $N = 16$; Experiment 3: $P = 0.015$, $d = 0.49$, $N = 19$; Experiment 4: $P = 0.078$, $d = 0.97$, $N = 6$; Fig. 4e–h). As expected, all subjects were more likely to choose an object when it was rewarded compared to when it was not rewarded on the previous trial (red bars in Fig. 4).

Finally, to further validate the models used for fitting, we simulated choice data using the estimated model parameters for individual subjects and computed the feature-based and object-based differential responses for the simulated data. The similarity between the average differential responses based on simulated and actual choice (dashed and solid lines in Fig. 4) indicated the success of our models in capturing behavior. Together, these results confirm the prediction of feature-based learning that reward values of all objects that share a feature with the selected object are updated based on the reward feedback.

**Influence of attention on feature-based learning.** Although the main goal of our study was to identify factors influencing how humans adopt feature-based vs. object-based learning, our design also allowed us to examine how attention may influence learning. In all our experiments, the two features of options provided different amounts of information. In principle, subjects could differentially attend to relevant features, resulting in assigning different weights to these features when learning or making decisions[16]. Therefore, we examined possible attentional effects by fitting choice behavior with a feature-based model with decay that has two separate learning rates for the less and more informative features (for subjects who adopted feature-based learning). By design, this model can also assign different weights to the two features. We expected that attention would result in a larger learning rate and/or weight for the more informative relative to the less informative feature. However, because the estimated learning rates and assigned weights for individual subjects were correlated and only their product captured the amount of change in choice behavior due to reward feedback, we used this product to measure difference between the two features. Moreover, we also computed feature-based differential response for the two features in order to compare how changes in the value of features of the object selected on the previous trial depend on how informative those features were.

First, we found a larger product of the learning rate and the assigned weight for the more informative than the less informative feature (one-sided sign-rank test; Experiment 1: $P = 0.044$, $d = 0.23$, $N = 38$; Experiment 2: $P = 0.031$, $d = 1.07$, $N = 5$; Experiment 3: $P = 0.0028$, $d = 0.90$, $N = 8$; Experiment 4: $P = 0.00075$, $d = 0.76$, $N = 19$; Fig. 5a–d). This indicates that subjects who adopted feature-based learning incorporated reward feedback (in terms of a combination of the learning rate and weights) from the more informative feature more strongly. Consistently, we found a larger feature-based differential response for the more informative feature in all experiments but this effect was not significant in Experiments 2 and 3 due to the small sample size (one-sided sign-rank test; Experiment 1: $P = 1.1 \times 10^{-4}$, $d = 0.72$, $N = 38$; Experiment 2: $P = 0.094$, $d = 0.89$, $N = 5$; Experiment 3: $P = 0.19$, $d = 0.20$, $N = 8$; Experiment

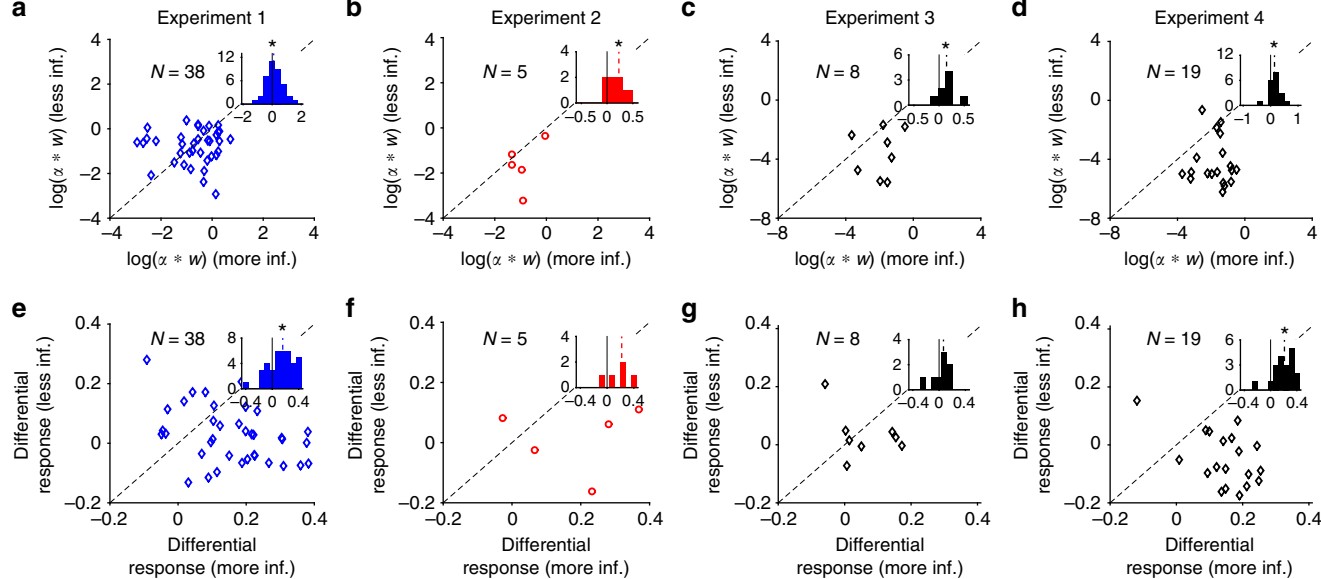

**Fig. 5** Feature-based learning was stronger for the more informative feature. **a–d** Plotted is the log product of the estimated learning rate ($\alpha$) and assigned weight ($w$) for the less informative feature (non-informative in the case of Experiments 2–4) vs. of the same product for the more informative feature for each individual, across four experiments. The insets show the histogram of difference in ($\alpha \times w$) between the more and less informative features. The dashed lines show the medians and the solid gray lines indicate zero. The star shows that the median of the difference in ($\alpha \times w$) were significantly different from 0 (one-sided sign-rank test, $P < 0.05$). These products were larger for the more informative feature in all experiments. **e–h** Plotted is the feature-based differential response for the less informative feature vs. the more informative feature. Conventions are the same as in **a–d**. The feature-based differential response was larger for the more informative feature in all experiments (though it did not achieve significance in Experiments 2 and 3), indicating that subjects updated their behavior more strongly for the more informative feature

4:
$P = 4.2 \times 10^{-5}$, $d = 1.2$, $N = 19$; Fig. 5e–h). This result illustrates that subjects updated their behavior to a greater extent for the more informative feature. The qualitative match between the analysis based on the product of estimated learning rates and feature weights and the analysis based on the differential response demonstrates the usefulness of model fitting in revealing important aspects of learning in our experiments. Overall, these results indicate that in dynamic or high-dimensional environments, subjects' choice behavior and learning were more strongly influenced by the information in the more informative feature, which could be due to deployment of attention on this feature.

To summarize our experimental results, we found that human subjects adopted feature-based learning in dynamic environments even when this approach did not reduce dimensionality. Subjects switched to object-based learning when the combination of features' values could not accurately predict all objects' values due to the lack of generalizable rules. Finally, in low-dimensional, static environments without generalizable rules, subjects still adopted feature-based learning first before gradually adopting object-based learning. Overall, these results demonstrate that feature-based learning might be adopted mainly to improve adaptability without compromising precision.

**Plausible mechanisms for learning and model adoption**. To understand neural mechanisms underlying model adoption in a multi-dimensional decision-making task, we examined two alternative network models that could perform such tasks (Fig. 6a, b). Because of their architectures, we refer to these models as the parallel decision-making and learning (PDML) model and the hierarchical decision-making and learning (HDML) model. Both models have two sets of value-encoding neurons that learn the reward values of individual objects (object-value-encoding neurons, OVE) or features (feature-value-

encoding neurons, FVE). Learning occurs in the synapses onto the value-encoding neurons that undergo reward-dependent plasticity, enabling these neurons to represent and update the values of presented objects or their features (see Methods section for more details). Despite this common rule of synaptic plasticity, there are many ways to combine signals from the OVE and FVE neurons and adjust the influence of these neurons on the final choice (arbitration mechanism). The PDML model makes two additional decisions using the output of an individual set of value-encoding neurons (OVE or FVE) which are then compared with the choice of the final decision-making (DM) circuit, which is based on the combination of output from both OVE and FVE neurons (Fig. 6a). If the final choice is rewarded (not rewarded), the model increases (decreases) the strength of connections between the set (or sets) of value-encoding neurons that produced the same choice as the final choice and the final decision-making circuit. This increases or decreases the influence of the set of value-encoding neurons that was more likely responsible for making the final correct or incorrect choice, respectively. By contrast, the HDML model utilizes a signal-selection circuit to determine which set of the value-encoding neurons contains a stronger signal, and updates connections from the OVE and FVE neurons to their corresponding signal-selection accordingly. In this model, signal strength is defined as the difference between the reward values of the two options based on the output of OVE or FVE neurons. The model uses only the output of the set with a stronger signal to make the final decision on a given trial (Fig. 6b). Subsequently, only the strength of connections between the set of value-encoding neurons producing the 'selected' signal and the corresponding neurons in the signal-selection circuit is increased or decreased depending on whether the final choice was rewarded or not rewarded, respectively (see Methods section for more details).

To show how these two different arbitration and learning mechanisms work, we first examined the simulated behavior of

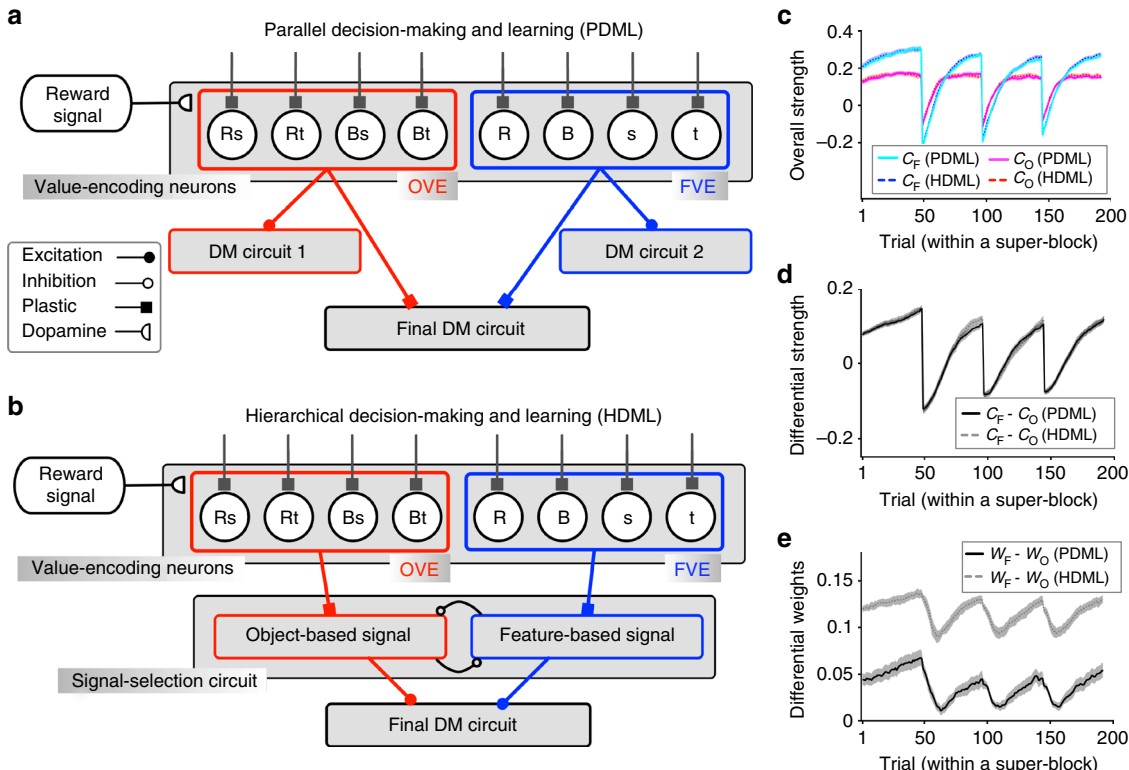

**Fig. 6** Architectures and performances of two alternative network models for multi-dimensional, decision-making tasks. **a**, **b** Architectures of the PDML (**a**) and the HDML (**b**) models. In both models, there are two sets of value-encoding neurons that estimate reward values of individual objects (object-value-encoding neurons, OVE) and features (feature-value-encoding neurons, FVE). The two models are different in how they combine signals from the OVE and FVE neurons and how the influence of these signals on the final decision is adjusted through reward-dependent plasticity. **c** The time course of the overall strengths of plastic synapses between OVE and FVE neurons and the final DM circuit ($C_O$ and $C_F$) in the PDML model, or between OVE and FVE neurons and the signal-selection circuit ($C_O$ and $C_F$) in the HDML model. These simulations were done for the generalizable environment (Experiment 1) where the block length was 48. **d** The difference between the $C_F$ and $C_O$ over time in the two models. **e** The difference in the overall weights of the two sets of value-encoding neurons on the final decision ($W_F$–$W_O$) for the same set of simulations shown in **c**, **d**

the two models during Experiment 1. The strength of connections from the OVE and FVE neurons to the final DM circuit in the PDML model or to the signal-selection circuit in the HDML model increased initially but at a much faster rate for FVE neurons (Fig. 6c). This occurred because on each trial, both features of a selected object were updated and thus, synapses onto FVE neurons were updated twice as frequently as those onto OVE neurons. These faster updates enabled the FVE neurons to signal a correct response more often than the OVE neurons following each change in reward probabilities (Fig. 6d). We also computed the overall weight of the feature-based and object-based approaches on the final choice, $W_F$ and $W_O$, respectively (Methods section). The difference between these two weights, ($W_F$–$W_O$), was positive in both models even though it decreased after each reversal, indicating that both models assigned a larger weight to feature-based than to object-based reward values. However, this effect was greater in the HDML than in the PDML model (Fig. 6e).

Next, we examined how well the PDML and HDML models can account for the observed choice behaviors in our experiments by simulating the behaviors in all four experiments using these models and analyzing the simulated and experimental data similarly. For Experiment 1, the simulated data using the PDML model were equally fit by the feature-based and object-based models (Fig. 7a) indicating that PDML could not adopt feature-based learning in a volatile, generalizable environment. In contrast, choice behavior of the PDML model during Experiment 2 was better fit by the object-based model similarly to the

experimental data (Fig. 7a). Choice behavior of the HDML model was consistent with our results in both Experiments 1 and 2 (Fig. 7f). For Experiment 3, both models adopted feature-based learning first and slowly transitioned to object-based learning (Fig. 7b, g) and, moreover, their choice behavior was better fit by object-based learning later in the experiment (Fig. 7c, h). Both models also adopted feature-based learning first during Experiment 4 but showed only a small transition toward object-based learning (Fig. 7d, i) such that their choice behavior was still better fit by feature-based learning even toward the end of the experiment (Fig. 7e, j). These patterns were consistent with those of experimental data during Experiments 3 and 4. Overall, choice behavior of the HDML model qualitatively matched the pattern of data in all experiments whereas the PDML model failed to capture choice behavior during Experiment 1.

We also tested the overall performance and the ability of HDML and PDML in adopting feature-based vs. object-based approach in a large set of environments, and examined how interactions between generalizability, frequency of changes in reward probabilities (volatility), and dimensionality affect the behavior of these models (Supplementary Note 1 and Supplementary Figs. 9 and 10). Overall, these simulations and accompanying analyses revealed that although both models were able to perform the task successfully, the HDML model exhibited higher performance and stronger adjustment of connections from the value-encoding neurons to the next level of computation. That is, the HDML was overall more successful in assigning more graded weights to different learning approaches according to

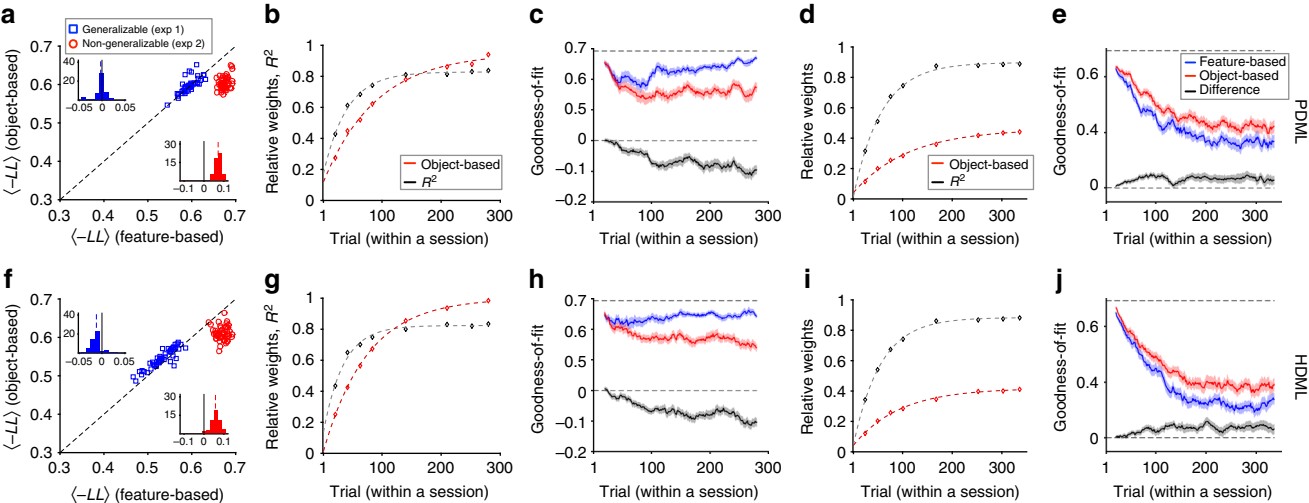

**Fig. 7** Replicating the pattern of experimental data using the PDML and HDML models. **a** Comparison of the goodness-of-fit in for the data generated by the PDML model in Experiments 1 (generalizable) and 2 (non-generalizable) using the object-based and feature-based RL models with decays. The insets show histograms of the difference in the negative log likelihood (-LL) based on the fits of the two models. In contrast to the experimental data, choice behavior of the PDML model in Experiment 1 was equally fit by the object-based and feature-based models. **b** The time course of model adoption in the PDML model. Plotted is the relative weight of object-based to the sum of the object-based and feature-based weights, and explained variance in estimates ($R^2$) over time in Experiment 3. Dotted lines show the fit of data based on an exponential function. **c** Transition from feature-based to object-based learning in the PDML model. Plotted are the average negative log likelihood based on the best feature-based model, best object-based RL model, and the difference between object-based and feature-based models in Experiment 3. Shaded areas indicate s.e.m., and the dashed line shows the measure for chance prediction. **d**, **e** The same as in **b**, **c**, but for simulations of Experiment 4. **f–j** The same as in **a–e**, but for the HDML model. Although both models qualitatively replicated the pattern of experimental data in Experiments 2–4, only the behavior of HDML model was consistent with data in Experiment 1

reward statistics in the environment. Therefore, hierarchical decision-making and learning might be more advantageous for adopting the model for learning in dynamic, multi-dimensional environments.

## Discussion

The framework proposed in this study for learning reward values in dynamic, multi-dimensional environments provides specific predictions about different factors that influence how humans adopt feature-based vs. object-based learning to tackle the curse of dimensionality. Our experimental results confirmed these predictions and demonstrated that dynamic environments tend to favor feature-based learning because this learning not only reduces dimensionality but also improves adaptability without compromising precision. When precision is compromised due to non-generalizability of the rules assumed for feature-based learning, object-based learning is adopted more frequently. Importantly, feature-based learning is initially adopted, even in the presence of non-generalizable rules that only slightly reduce dimensionality and when reward contingencies do not change over time. These results suggest that the main driver for adopting heuristic feature-based learning is increasing adaptability without compromising precision; that is, to overcome the adaptability-precision tradeoff (APT).

The APT sets an important constraint on learning reward values in a dynamic environment where they change over time. One solution to mitigate the APT is to adjust learning over time via metaplasticity[14, 15]. Nevertheless, even with adjustable learning, the APT still persists and becomes more critical in multi-dimensional environments, since the learner may never receive reward feedback on many unchosen options and feedback on chosen options is limited. Importantly, adopting heuristic feature-based learning enables more updates after each reward feedback, which can enhance the speed of learning without adding noise, as with other heuristic learning mechanisms[17]. Moreover, such

learning allows estimation of reward values for options which have never been encountered before[18, 19].

Our results could explain why learning in young children, which is limited by the small number of feedback, is dominated by attending to individual features (e.g., choosing a favorite color) to such an extent that it prevents them from performing well in simple tasks such as the dimension-switching task[20]. Interestingly, this inability has been attributed to failing to inhibit attention to the previously relevant or rewarding feature[21]. Here we propose an alternative possibility that by focusing on a single feature such as color, children could evaluate reward outcomes of chosen options based on color and thus increase their learning speed. Moreover, by choosing a favorite color, they can further reduce dimensionality by decreasing the number of feature instances/categories to just two: favorite and non-favorite color. Thus, our results explain that choosing a favorite color not only reduces dimensionality but also increases adaptability without compromising precision.

Although rules used for the heuristic feature-based approach are only partially generalizable in the real world, this lack of generalizability may not prevent humans from adopting feature-based learning for a few reasons. First, simply due to chance, the level of generalizability is larger for a higher dimensionality if there is at least one informative feature in the environment. Second, reward values of features can be learned separately for different domains (e.g., color of fruits and color of cars). Thus, the actual values of dimensionality and generalizability in the real world depend on how an individual separates learning for different domains. Finally, it might be practically difficult to detect non-generalizability due to a very large number of features and options (or domains of learning) in the real world. Accordingly, feature-based learning could provide a "fast and frugal way" for learning in the real world[22].

Heuristic feature-based learning is computationally less expensive and more feasible than object-based learning, since it can be achieved using a small number of value-encoding neurons

with pure feature selectivity, namely neurons that represent the reward value in a single dimension, such as color or shape. By comparison, object-based learning requires myriad-mixed selectivity neurons tuned to specific combinations of various features. Thus, in contrast to recent theoretical work that has highlighted the advantage and importance of non-linear, mixed selectivity representation for cognitive functions[23, 24], our work points to the importance of pure feature selectivity for reward representation. The advantage of mixed selectivity representation could be specific to tasks with low dimensionality (in terms of reward structure) or when information does not change over time such as in object categorization tasks[25–28].

Our computational and experimental results also provide a few novel neural predictions. First, they predict that learning about reward in dynamic environments could depend more strongly on value-encoding neurons with pure feature selectivity, since activity or representation of such neurons can be adjusted more frequently over time due to more updates per feedback. Second, considering that neurons with pure feature selectivity are also crucial for saliency computations[29], modulations of these neurons by reward could provide an effective mechanism for the modulation of attentional selection by reward[30, 31]. Third, they predict larger learning rates for neurons with highly mixed selectivity; otherwise, the information in these neurons would lag the information in pure feature-selective neurons and become obsolete. Fourth, the complexity of reward value representation should be directly related to the stability of reward information in the environment. As the environment becomes more stable, learning the reward value of conjunctions of features and objects becomes more feasible and thus, more complex representation of reward values will emerge. Finally, updates of reward values for different objects and features necessitate separate reward prediction error (RPE) signals. This predicts different roles for multiple RPE signals observed in areas other than striatum and midbrain dopaminergic system (substantia nigra and ventral tegmental area), such as the anterior cingulate and medial prefrontal cortices (see ref. [32] for a review). These novel predictions could be tested in future experiments.

Our framework for understanding model adoption has a few limitations. First, it does not address the influence of what has been learned on learning strategy and future model adoption[31]. Second, it does not determine the intermediate steps for transition from feature-based to object-based learning, for example, what "conjunctions" of features are constructed and learned over time. Third, it does not address how model adoption depends on the ability of the decision maker to discriminate between feature values (e.g., colors of objects). Future model development and experiments are required to explore these questions.

As our computational modeling suggests that learning based on feature-based and object-based approaches occurs simultaneously in two separate circuits, arbitration between the two forms of learning might be required[33, 34]. Our modeling results show that such arbitration could happen via competition between two circuits based on the strength of signals in each circuit. Although we could not directly fit experimental data using these two models due to the over-fitting problem and large between-subject variability, our experimental results are qualitatively more compatible with a hierarchical decision-making and learning (HDML) model, since the parallel decision-making and learning model does not show the sensitivity to experimental factors observed in our human subjects. In the HDML model, the best sources of information were identified to make decisions, and weights associated with the selected sources were successively updated according to reward feedback. The hierarchical structure allows the HDML model to reduce noise in decision making by ignoring the less informative value-coding network on each trial.

Our results imply that reward feedback alone can correctly adjust behavior toward a more object-based or a more feature-based approach, without any explicit optimization or knowledge of the environment. This does not contradict the need for arbitration but instead provides a simple mechanism for arbitration based on the same reward feedback used to learn reward values of objects and features. Interestingly, competition through stages of hierarchy has also been suggested as an underlying mechanism behind multi-attribute decision making[35, 36]. The HDML model proposed in this study shares some components with the model of Hunt et al. (2014)[35], namely competition between different attribute values before they are combined, though our model includes learning as well. Similar to Wunderlich et al. (2011)[7], we also suggest that the brain should learn values and weights for all possible informative dimensions and update these weights on every trial.

Despite the fact that naturalistic learning from reward feedback entails options with overlapping features, only recently have some studies used multi-dimensional experimental paradigms to study learning from reward feedback and explored possible solutions for the curse of dimensionality[5–7, 35, 37, 38]. A few of these studies have found that learning in a multi-dimensional environment relies on constructing a simplified representation of the stimuli via attending to one feature and ignoring others[5, 37]. Similarly, we found the evidence for attentional bias on one of the two features, whereas no feature was completely ignored. Interestingly, a recent study has shown that attention, guided by ongoing learning, can bias both value computation and update, and thus results in more efficient learning in multi-dimensional environments[16]. Finally, attending to only a subset of "relevant" features is both inevitable and crucial for learning and decision making in high-dimensional environments[39, 40]. However, in order to identify the relevant features in dynamic environments, values of multiple features should be updated in parallel over time.

In conclusion, we show that a tradeoff between adaptability and precision could explain why humans adopt feature-based learning, especially in dynamic environments. Moreover, our results suggest that neurons with pure selectivity could be crucial for learning in dynamic environments and could provide a missing framework for understanding how heterogeneity in reward representation emerges[41, 42].

## Methods

**Framework for adoption of feature-based vs. object-based learning.** We first developed a general framework for understanding model adoption during learning in dynamic, multi-dimensional environments. The decision maker's task is to learn the reward values of a set of options via reward feedback after selecting one of two alternative options on each trial. We simulated the behavior of two contrasting learners in this task: object-based and feature-based learners. The object-based learner directly estimates the value of individual objects via reward feedback. By contrast, the feature-based learner estimates the reward values of all feature instances (e.g., red, blue, square, or triangle) by updating the reward values associated with all the features of the object for which reward feedback is given. This learner then combines the reward values of feature instances to estimate the overall reward value of individual objects.

Assuming that options/objects have $m$ features, each of which can have $n$ different instances, there are $n^m$ possible objects in the environment. For example, if an object has two features ($m = 2$), color and shape, and there are three colors and three shapes ($n = 3$), there would be nine ($3^2$) possible objects in the environment. We first constructed an environment by assigning a probability of reward to each object based on the feature instances of that object. More specifically, $n$ feature instances were assigned with a set of equally-spaced (in log scale) odds ratios (OR) in all $m$ dimensions. The minimum and maximum values of ORs were set to $1/x$ and to $x$ ($x > 1$), respectively. For example, for $n = 3$, OR $(F_{ij}) = \{1/2, 1, 2\}$ where $F_{ij}$ is the feature instance $j$ ($j = 1, ..., n$) of feature $i$ ($i = 1, ..., m$). The OR for a given object $a$, $\text{OR}(O_a)$, was determined by multiplying the ORs of all features of that object: $\text{OR}(O_a) = \prod_{i=1, \text{ for } F_{ij} \text{ present in } O_a}^{m} \text{OR}(F_{ij})$. Finally, the probability of reward on each object was then computed by transforming the object's LR to the probability of reward: $p_r(O_a) = \text{OR}(O_a)/(1 + \text{OR}(O_a))$. These reward probabilities are referred to collectively as the fully generalizable reward matrix.

Although each object is assigned a reward value, the feature-based learner could instead use the reward value for each feature instance (e.g., red, blue, triangles, and squares) to estimate reward values of objects in two steps. First, the reward value for a given feature instance (e.g., red) can be computed by averaging the reward values of all objects that contain that feature instance (e.g., all red objects): $\overline{p_r}(F_{ij}) = (1/n^{m-1}) \sum_{O_a \text{ contains } F_{ij}} p_r(O_a)$. Second, an estimated reward value, $\tilde{p}_r(O_a)$, can be generated by combining the reward values of features using the Bayes theorem:

$$\tilde{p}_r(O_a) = \left(\overline{p_r}(F_{1j}) \times \overline{p_r}(F_{2k}) \times \ldots \right) / \left(\overline{p_r}(F_{1j}) \times \overline{p_r}(F_{2k}) \times \ldots \right.$$
$$\left. + (1 - \overline{p_r}(F_{1j})) \times (1 - \overline{p_r}(F_{2k})) \times \ldots \right) \quad (1)$$
$$\text{for } O_a \text{ containing of } F_{1j}, F_{2k}, \text{ etc.}$$

These estimated reward probabilities constitute the estimated reward matrix based on features. The rank order of probabilities in the estimated reward matrix, which determines preference between objects, is similar to that of the fully generalizable reward matrix, whereas the exact values may differ slightly (diamonds in Supplementary Fig. 1).

By randomly shuffling the elements of the fully generalizable reward matrix in all feature dimensions except one, which we call the informative feature, we generated environments with different levels of generalizability. We used the correlation between the 'shuffled' reward matrix and estimated reward matrix (i.e., correlation between the actual reward value of options and the estimated reward value of options based on their features) to define a generalizability index. On the basis of this definition, the generalizability index can take on any value between −1 and 1. Without any shuffling, we get a fully generalizable environment (generalizability index equal to 1) where the rank order of the estimated reward values of options based on their features is identical to the rank order of actual objects' values. With less generalizability, the rank order is not the same (reflected in a smaller generalizability index) and the difference between the estimated reward values based on features and actual objects' values increases (Supplementary Fig. 1).

The task for the decision maker is to learn the reward value of options/objects via reward feedback in order to choose between two alternative options in each trial. To illustrate how learning strategy influences performance in this task, we considered two alternative learners: object-based and feature-based. We assumed that the object-based learner directly estimates the reward value of all objects using reward feedback in each trial based on the following equations:

$$V_{O_a}(t+1) = V_{O_a}(t) + \alpha(1 - V_{O_a}(t)), \text{ if } r(t) = 1$$
$$V_{O_a}(t+1) = V_{O_a}(t) - \alpha(V_{O_a}(t)), \text{ if } r(t) = 0 \quad (2)$$

where $t$ represents the trial number, $V_{O_a}(t)$ is the reward value of the chosen object $a$, $r(t)$ is the trial outcome (1 for rewarded, 0 for unrewarded), and $\alpha$ is the learning rate. The value of the unchosen object is not updated. By contrast, the feature-based learner estimates the reward value of individual feature instances (e.g., red, blue, triangles, and quares), $V_{F_{ij}}(t)$, using the same update rule as in Equation 2, but applying to all features of the chosen object. This learner then combines the reward values of feature instances to compute reward values of each option (Eq. 1). Therefore, the object-based learner only updates one value function after each feedback, whereas the feature-based learner updates the reward value of all feature instances of the selected object.

To measure how well a learner that uses the object-based approach can differentiate between different options at a given point in time, we defined the differential signal, $S_O(t)$, in the object-based learning model as follows:

$$S_O(t) = \frac{1}{n^m \times (n^m - 1)} \sum_{a=1}^{n^m} \sum_{b=1}^{n^m} (V_{O_a}(t) - V_{O_b}(t)) \text{sign}(p_r(O_a) - p_r(O_b)) \quad (3)$$

where $p_r(O_a)$ is the probability of reward on object $a$. The differential signal for the feature-based learning model, $S_F(t)$, was computed by replacing $V_{O_a}(t)$ in the above equation with the estimated reward value $\tilde{V}_{O_a}(t)$, which was computed by replacing $\overline{p_r}(F_{ij})$ in Equation 1 with $V_{F_{ij}}(t)$. Therefore, the differential signal measures how reward values estimated by a given model correctly differentiate between actual reward values of objects.

By comparing the time courses of the differential signal for the object-based and feature-based learners (using the same learning rate and similar initial conditions), we computed the time at which the object-based learner carries a stronger differential signal than the feature-based learner (the 'cross-over point'). A larger cross-over point indicates the superiority (better performance) of feature-based relative to object-based learning for a longer amount of time, whereas a zero cross-over point indicates that object-based learning is always superior.

**Subjects**. Subjects were recruited from the Dartmouth College student population (ages 18–22 years). In total, 59 subjects were recruited (34 females) to perform the choice task in Experiment 1 and/or 2 (18 in Experiment 1 only, 8 in Experiment 2 only, and 33 in both experiments). This produced behavioral data from 51 and 41 subjects for Experiments 1 and 2, respectively. A general linear model for predicting performance did not reveal any effects of previous participation or the

order of experiments for those who performed in both Experiments 1 and 2. To exclude subjects whose performance was not significantly different from chance (0.5), we used a performance threshold of 0.5406 (equal to 0.5 plus 2 times s.e.m., based on the average of 608 trials after excluding the first 10 trials of each block in Experiment 1 or 2). This resulted in the exclusion of data from 8 of 51 subjects in Experiment 1, and 19 of 41 subjects in Experiment 2. An additional subject was excluded from Experiment 2 for submitting the same response throughout the entire experiment. The remaining 64 datasets ($N = 43$ and 21 for Experiments 1 and 2, respectively) were used for the main analyses but the excluded data sets were analyzed as well. For Experiment 3, 36 additional subjects were recruited (20 females) and a performance threshold of 0.5447 (equal to 0.5 plus 2 times s.e.m., based on the average of 500 trials after excluding the first 30 trials of each session) was used to exclude subjects whose performance was indistinguishable from chance ($N = 9$). In total, only two subjects participated in all three experiments, and this occurred over 4 months. For Experiment 4, 36 new subjects were recruited (22 females) and a performance threshold of 0.5404 (equal to 0.5 plus 2 times s.e.m., based on the average of 612 trials after excluding the first 30 trials of each session) was used to exclude subjects whose performance was indistinguishable from chance ($N = 11$). No subject had a history of neurological or psychiatric illness. Subjects were compensated with a combination of money and "t-points," which are extra-credit points for classes within the Department of Psychological and Brain Sciences at Dartmouth College. The base rate for compensation was \$10/h or 1 t-point/h. Subjects were then additionally rewarded based on their performance by up to \$10/h. All experimental procedures were approved by the Dartmouth College Institutional Review Board, and informed consent was obtained from all subjects before participating in the experiment. Finally, all experiments were written in MATLAB, using the Psychophysics Toolbox Version 3 extensions[43] and presented using an OLED monitor.

**Experiments 1 and 2**. In each of these experiments, subjects completed two sessions (each session composed of 384 trials and lasting about half an hour) of a choice task during which they selected between a pair of objects on each trial (Supplementary Fig. 2a). Objects were one of four colored shapes: blue triangle, red triangle, blue square, and red square. Subjects were asked to choose the object that was more likely to provide a reward in order to maximize the total number of reward points, which would be converted to monetary reward and/or t-points at the end of the experiment.

In each trial, the selection of an object was rewarded only according to its reward probability and independently of the reward probability of the other object. This reward schedule was fixed for a block of trials (block length, $L = 48$), after which it changed to another reward schedule without any signal to the subject. Sixteen different reward schedules consisting of some permutations of four reward probabilities, (0.1, 0.3, 0.7, and 0.9), were used. In eight of these schedules, a generalizable rule could be used to predict reward probabilities for all objects based on the combinations of their feature values (Supplementary Fig. 2b). In the other eight schedules, no generalizable rule could be used to predict reward probabilities for all objects based on the combinations of their feature values (Supplementary Fig. 2c). For example, the schedule notated as 'Rs' indicates that red objects are much more rewarding than blue objects, square objects are more rewarding than triangle objects, and color (uppercase 'R') is more informative than shape (lower case 's'). In this generalizable schedule, red square was the most rewarding object whereas blue triangle was the least rewarding object. For non-generalizable schedules, only one of the two features was on average informative of reward values. For example, the 'r1' schedule indicated that, overall, red objects were slightly more rewarding than blue objects, but there was no generalizable relationship between the reward values of individual objects and their features (e.g., red square was the most rewarding object, but red triangle was less rewarding than blue triangle). In other words, the non-generalizable reward schedules were designed so that a rule based on feature combination could not predict reward probability on all objects. For example, learning something about a red triangle did not necessarily tell the subject anything about other red objects or other triangle objects.

The main difference between Experiments 1 and 2 was that their environments were composed of reward schedules with generalizable and non-generalizable rules, respectively (Supplementary Fig. 2d, f). In both experiments, as the subjects moved between blocks of trials, reward probabilities for the informative features were reversed without any changes in the average reward probabilities for the less informative and non-informative feature in Experiments 1 and 2, respectively. For example, going from Rs to Bs changes the more informative feature instance from red to blue. Reward probabilities changed without any cue to the subject and created dynamic environments. In addition, the average reward probabilities for the less informative or non-informative feature changed (e.g., from Bs and Rs to Bt and Rt) every four blocks (super-blocks; Supplementary Fig. 2e, g). Each subject performed the experiment in each environment once, where either color or shape was consistently more informative. The more informative feature was randomly assigned and counter-balanced across subjects to minimize the effects of intrinsic color or shape biases. The order of experiments was randomized for subjects who performed both Experiments 1 and 2.

**Experiment 3**. In this experiment, subjects completed two sessions, each of which included 280 choice trials interleaved with five or eight short blocks of estimation

trials (each block with eight trials). On each trial of the choice task, the subject was presented with a pair of objects and was asked to choose the object that they believed would provide the most reward. These objects were drawn from a set of eight objects, which were constructed using combinations of three distinct patterns and three distinct shapes (Supplementary Fig. 8a; one of nine possible objects with a reward probability of 0.5 was excluded to shorten the duration of the experiment). The three patterns and shapes were selected randomly for each subject from a total of eight patterns and eight shapes. The two objects presented on each trial always differed in both pattern and shape. Other aspects of the choice task were similar to those in Experiments 1 and 2, except that reward feedback was given for both objects rather than just the chosen object, in order to accelerate learning. During estimation blocks, subjects provided their estimates of the probability of reward for individual objects. Possible values for these estimates were from 5 to 95%, in 10% increments (Supplementary Fig. 8c). All subjects completed five blocks of estimation trials throughout the task (after trials 42, 84, 140, 210, and 280 of the choice task), and some subjects had three additional blocks of estimation trials (after trials 21, 63, and 252) to better assess the estimations over time. Each session of the experiment was about 45 min in length, with a break before the beginning of the second session. The second session was similar to the first, but with different sets of shapes and patterns.

Selection of a given object was rewarded (independently of the other presented object) based on a reward schedule with a moderate level of generalizability such that reward probability of some individual objects could not be determined by combining the reward values of their features. Because of the larger number of objects, the reward schedule was more complex than that used in Experiment 1, but did not change over the course of the experiment. Non-generalizable reward matrices can be constructed in many ways. In Experiment 3, one feature (shape or pattern) was informative about reward probability while the other was not. Although the informative feature (e.g., pattern and shape in left and right panels of Supplementary Fig. 8a, respectively) was on average predictive of reward, this prediction was not generalizable. That is, some objects that contained the most rewarding feature instances were still less rewarding than objects that did not contain these feature instances. For example, S1P3 in the left panel of Supplementary Fig. 8a was less rewarding than S1P2. Finally, the average reward probability of the objects with the same non-informative feature instances (e.g., S1P1, S1P2, and S1P3 in Supplementary Fig. 8a, left panel) was always 0.5. This reward schedule ensured that subjects would not be able to predict reward probability accurately for all objects based on the combination of their feature values. Similar to Experiments 1 and 2, the informative feature was randomly assigned and counter-balanced across subjects to minimize the effects of intrinsic pattern or shape biases.

**Experiment 4**. This experiment was similar to Experiment 3, except that we used four feature instances for each feature (shape and pattern) resulting in an environment with a higher dimensionality. Each subject completed two sessions, each of which included 336 choice trials interleaved with five or eight short blocks of estimation trials (each block with eight trials). The objects in this experiment were drawn from a set of twelve objects, which were combinations of four distinct patterns and four distinct shapes (Supplementary Fig. 8b; four of sixteen possible objects with reward probability 0.5 were removed to shorten the duration of the experiment). The four patterns and shapes were selected randomly for each subject. The probabilities of reward on different objects (reward matrix) were set such that there was one informative feature, and the minimum and maximum average reward values for features were similar for Experiments 3 and 4.

**Data analysis**. We utilized the information subjects provided during estimation trials of Experiments 3 and 4 to examine how they determined the reward values of objects using two alternative methods. First, we used linear regression to fit the estimates of reward probabilities as a function of the following variables: actual reward probabilities assigned to each object (object-based term); the reward probabilities estimated based on the combination of the reward values of features (feature-based term) using the Bayes theorem (Eq. 1); and a constant. The constant (bias) in this regression model quantifies subjects' overall bias in reporting probabilities. Second, to determine whether subjects' estimates were closer to estimates based on the feature-based or object-based approach, we computed the correlation between subjects' estimates and the actual reward probabilities assigned to each object, or subjects' estimates and the reward probabilities estimated using the reward values of features (Eq. 1). Unless otherwise mentioned, the statistical comparisons were performed using Wilcoxon signed rank test in order to test the hypothesis of zero median for one sample or the difference between paired samples. The reported effect sizes are Cohen's d values. All behavioral analyses, model fitting, and simulations were done using MATLAB 2015a (MathWorks, Inc., Natick, MA).

**Testing the behavioral predictions of feature-based learning**. To measure the direct effect of feature-based learning on choice behavior, we defined the feature-based 'differential response' to reward feedback equal to the difference between the probability of selecting an object that contains only one of the two features of the object selected and rewarded on the previous trial (i.e., rewarded object) when

this object was paired with an object that did not share any feature with the previously rewarded object minus the same probability when the previous trial was not rewarded. For example, denoting the probability of choosing X when it is presented together with Y in trial $t$ given that choosing Z was rewarded and unrewarded in the previous trial as $p(\text{X>Y}(t)|\text{ Z}(t-1)+)$ and $p(\text{X>Y}(t)|\text{ Z}(t-1)-)$, $p(\square > \triangle (t)|\blacksquare(t-1)+) - p(\square > \triangle(t)|\blacksquare(t-1)-)$ measures the change in the value of features of the object selected on the previous trial. For comparison, we also calculated the object-based differential response to reward feedback equal to the difference between the probability of selecting the object that was selected and rewarded on the previous trial and the same probability when the previous trial was not rewarded (e.g., $p(\text{choose }\blacksquare$ in trial $t|\blacksquare(t-1)+) - p(\text{choose }\blacksquare$ in trial $t|\blacksquare(t-1)-))$. The object-based differential response measures the change in preference for an object due to reward feedback and is equivalent to the difference between win-stay and lose-stay strategy for a particular object.

**Model fitting procedure**. To capture subjects' learning and choice behavior, we used seven different reinforcement learning (RL) models based on object-based or feature-based approaches. These models were fit to experimental data by minimizing the negative log likelihood of the predicted choice probability given different model parameters using the 'fminsearch' function in MATLAB (MathWorks, Inc.). To avoid finding local minima for the fit of experimental data or simulated choice behavior, we repeated fitting of each dataset with at least 10 different initial conditions and picked the best fit among all those fits. Based on the examination of the outcome fits, we found that 10 initialization to be enough to avoid local minima. We computed three measures of goodness-of-fit in order to determine the best model to account for the behavior in each experiment: average negative log likelihood (-LL), Akaike information criterion (AIC), and Bayesian information criterion (BIC). The smaller value for each measure indicates a better fit of choice behavior.

**Object-based RL models**. In this group of models, the reward value of each object is directly estimated from reward feedback on each trial using a standard RL model[4]. For example, in the uncoupled object-based RL, only the reward value of the chosen object is updated on each trial. This update is done via separate learning rates for rewarded or unrewarded trials using the following equations, respectively[42]:

$$V_{\text{choO}}(t + 1) = V_{\text{choO}}(t) + \alpha_{\text{rew}}(1 - V_{\text{choO}}(t)), \text{if } r(t) = 1$$
$$V_{\text{choO}}(t + 1) = V_{\text{choO}}(t) - \alpha_{\text{unr}}(V_{\text{choO}}(t)), \text{if } r(t) = 0 \quad (4)$$

where $t$ represents the trial number, $V_{\text{choO}}$ is the estimated reward value of the chosen object, $r(t)$ is the trial outcome (1 for rewarded, 0 for unrewarded), and $\alpha_{\text{rew}}$ and $\alpha_{\text{unr}}$ are the learning rates for rewarded and unrewarded trials. The value of the unchosen object is not updated in this model.

In the coupled object-based RL, the reward values of both objects presented on a given trial are updated, but in opposite directions (assuming that reward assignments on the two objects are anti-correlated). That is, while the value of chosen object is updated based on Equation 4, the value of unchosen object is updated based on the following equation:

$$V_{\text{uncO}}(t + 1) = V_{\text{uncO}}(t) - \alpha_{\text{rew}}(V_{\text{uncO}}(t)), \text{if } r(t) = 1$$
$$V_{\text{uncO}}(t + 1) = V_{\text{uncO}}(t) + \alpha_{\text{unr}}(1 - V_{\text{uncO}}(t)), \text{if } r(t) = 0 \quad (5)$$

where $t$ represents the trial number and $V_{\text{uncO}}$ is the estimated reward value of the unchosen object.

The estimated value functions are then used to compute the probability of selecting between the two objects on a given trial (O1 and O2) based on a logistic function:

$$\text{logit } P_{O1}(t) = (V_{O1}(t) - V_{O2}(t))/\sigma + \text{bias} \quad (6)$$

where $P_{O1}$ is the probability of choosing object 1, $V_{O1}$ and $V_{O2}$ are the reward values of the objects presented to the left and right, respectively, bias measures a response bias toward the left option to capture the subject's location bias, and $\sigma$ is a parameter measuring the level of stochasticity in the decision process.

**Feature-based RL models**. In this group of models, the reward value (probability) of each object is computed by combining the reward values of the features of that object, which are estimated from reward feedback using a standard RL model. The update rules for the feature-based RL models are identical to the object-based ones, except that the reward value of the chosen (unchosen) object is replaced by the reward values of the features of the chosen (unchosen) object. In Experiments 3 and 4, the two alternative objects were always different in both features. In Experiments 1 and 2, however, the two alternative objects could have a common feature instance (e.g. both are blue) and updating the reward value of this common feature could be problematic. Indeed, we found that the fit of choice behavior based on a feature-based model which always updates the reward values of both features of the selected object on each trial was worse than that of all other tested models (data not shown). Therefore, in the feature-based models presented here, only the

reward value of the unique feature is updated when the two alternative options have a common feature on a given trial.

As with the object-based RL models, the probability of choosing an object is determined based on the logistic function of the difference between the estimated values for the objects presented

$$\text{logit } P_{O1}(t) = w_{\text{shape}}\left(V_{\text{shapeO1}}(t) - V_{\text{shapeO2}}(t)\right) + w_{\text{color}}(V_{\text{colorO1}}(t) - V_{\text{colorO2}}(t)) + \text{bias}$$
(7)

where $V_{\text{shapeO1}}$ ($V_{\text{colorO1}}$) and $V_{\text{shapeO2}}$ ($V_{\text{colorO2}}$) are the reward values associated with the shape (color) of left and right objects, respectively, bias measures a response bias toward the left option to capture the subject's location bias, and $w_{\text{shape}}$ and $w_{\text{color}}$ determine the influence of the two features on the final choice. Note that these weights can be assumed to be learned over time through reward feedback (as in our models; see below) or could reflect differential processing of the two features due to attention.

**RL models with decay**. Additionally, we investigated the effect of 'forgetting' the reward values of unchosen objects or feature(s) by introducing decay of value functions (in the uncoupled models), which have been shown to capture some aspects of learning[44, 45], especially in multi-dimensional tasks[5]. More specifically, the reward values of unchosen objects or feature(s) decay to 0.5 with a rate of $d$, as follows:

$$V(t+1) = V(t) - d \times (V(t) - 0.5)$$
(8)

where $t$ represents the trial number and $V$ is the estimated reward probability of an object or a feature.

**Validation of fitting procedure for detecting learning approach**. To investigate whether our fitting procedure can be used to distinguish between alternative models and to accurately estimate model parameters, we simulated the afore-mentioned six models over a range of parameters in the four experiments. The simulated data were generated using the learning rate ($q$) ranging from 0.05 to 0.4, the decay rate ($d$) ranging from 0.005 to 0.04, and the stochasticity in choice ($\sigma$) ranging from 0.05 to 0.4. Parameters outside these ranges did not result in an appropriate model behavior in our experiments. We then fit the simulated data with all of the models to compute the goodness-of-fit and to estimate model parameters. The goodness-of-fit and the error in the estimated model parameters (the absolute difference between the actual and estimates) were computed by averaging over all fits based on all sets of parameters.

**Estimating attentional effects**. Attention could influence how reward values of two features determine choice and how they are updated over time. Therefore, in order to distinguish these two roles of attention, we estimated learning rates as well as the 'attentional' weights separately for the less and more informative features. By design, the feature-based models assign two different weights to the two features before combining them to make a choice (Eq. 7). We also extended the feature-based model with decay to include separate learning rates for the less and more informative features. For fitting of choice behavior in Experiments 3 and 4, we adopted two sets of weights for the first and second session of the experiments since two different sets of stimuli were used in these two sessions.

**Computational models**. To gain insight into the neural mechanisms underlying multi-dimensional decision-making, we examined two possible network models that could perform such a task (Fig. 6a, b). Both models have two sets of value-encoding neurons that learn the reward values of individual objects (object-value-encoding neurons, OVE) or features (feature-value-encoding neurons, FVE). More specifically, plastic synapses onto value-encoding neurons undergo reward-dependent plasticity (via reward feedback), which enables these neurons to represent and update the values of presented objects or their features. Namely, reward values associated with individual objects and features are updated by potentiating or depressing plastic synapses onto neurons encoding the value of a chosen object or its features depending on whether the choice was rewarded or not rewarded, respectively.

The two network models differ in how they combine signals from the OVE and FVE neurons and how the influence of signals from these neurons on the final choice is adjusted based on reward feedback. More specifically, the parallel decision-making and learning (PDML) model makes two additional decisions using the output of an individual set of value-encoding neurons (OVE or FVE) and compares them with the choice of the final decision-making (DM) circuit, which is based on the combination of output from both OVE and FVE neurons(Fig. 6a). If the final choice is rewarded (not rewarded), the model increases (decreases) the strength of connections between the set or sets of value-encoding neurons that produced the same choice as the final choice. This increases or decreases the influence of the set of value-encoding neurons that was more likely responsible for making the final correct or incorrect choice, respectively. By contrast, the hierarchical decision-making and learning (HDML) model updates connections from the OVE and FVE neurons to the corresponding neurons in the signal-

selection circuit by determining which set of the value-encoding neurons contains a stronger signal (the difference between the values of the two options) first, and uses only the outputs of that set to make the final decision on a given trial (Fig. 6b). Subsequently, only the strengths of connections between the set of value-encoding neurons responsible for the 'selected' signal and the corresponding neurons in the signal-selection circuit are increased or decreased depending on whether the final choice was rewarded or not rewarded, respectively.

**Learning rule**. We assumed that plastic synapses undergo a stochastic, reward-dependent plasticity rule (see refs. [46,47] for details). Briefly, we assumed that plastic synapses are binary and could be in potentiated (strong) or depressed (weak) states. On every trial, plastic synapses undergo stochastic modifications (potentiation or depression) depending on the model's choice and reward outcome (see below). During potentiation events, a fraction of weak synapses transition to the strong state with probability $q_+$. During depression events, a fraction of strong synapses transition to the weak state with probability $q_-$. These modifications allow a given set of plastic synapses to estimate reward values associated with an object or feature[46–49].

For binary synapses, the fraction of plastic synapses that are in the strong state (which we call 'synaptic strength') determines the firing rate of afferent neurons. We denote the synaptic strength of plastic synapses onto a given population of value-encoding neurons 'v' by $F_v(t)$, where $v = \{R, B, s, t, Rs, Bs, Rt, Bt\}$ represents a pool of neurons encoding the value of a given feature or a combination of features (in Experiments 1 and 2), and $t$ represents the trial number. In Experiments 3 and 4, the number of feature instances was three and four, respectively, instead of two, resulting in six and eight sets of FVE neurons and nine and sixteen sets of OVE neurons, respectively. Similarly, we denote the synaptic strength of plastic synapses from value-encoding neurons to the final DM circuit in the PDML model, or to the signal-selection circuit in the HDML model, by $C_m(t)$ where $m = \{O, F\}$ represents general connections from OVE and FVE neurons, respectively.

The changes in the synaptic strengths for synapses onto value-encoding neurons depend on the model's choice and reward outcome on each trial. More specifically, we assumed that synapses selective to the chosen object or features of the chosen object undergo potentiation or depression depending on whether the choice was rewarded or not, respectively:

$$F_{v(\text{ch})}(t+1) = F_{v(\text{ch})}(t) + q_+\left(1 - F_{v(\text{ch})}(t)\right), \text{ if } r(t) = 1$$
$$F_{v(\text{ch})}(t+1) = F_{v(\text{ch})}(t) - q_- F_{v(\text{ch})}(t), \text{ if } r(t) = 0$$
(9)

where $t$ represents the trial number, $F_{v(\text{ch})}(t)$ is the synaptic strength for synapses selective to the chosen object or features of the chosen object, $r(t)$ is the reward outcome, and $q_+$ and $q_-$ are potentiation and depression rates, respectively. The rest of plastic synapses transition to the weak state, according the following equation

$$F_{v(\text{unch})}(t+1) = F_{v(\text{unch})}(t) - q_d \times \left(F_{v(\text{unch})}(t) - 0.5\right)$$
(10)

where $F_{v(\text{unch})}(t)$ is the synaptic strength for synapses selective to the unchosen object or features of the unchosen object, and $q_d$ is the depression rate for the rest of plastic synapses. Note that similarly to the models used for fitting, only the reward value of the unique feature of the selected object was updated when the two alternative objects had a common feature.

We used similar learning rules for plastic synapses from value-encoding neurons to the final DM circuit in the PDML model as we did from value-encoding neurons to the signal-selection circuit in the HDML model. In the PDML model, plastic synapses from value-encoding neurons to the final DM circuit are updated depending on additional decisions based on the signal in an individual set of value-encoding neurons (OVE or FVE), the final choice, and the reward outcome as follows:

$$C_m(t+1) = C_m(t) + q_+(1 - C_m(t)), \text{ if } r(t) = 1, \text{ and pool } m \text{ choice} = \text{final choice}$$
$$C_m(t+1) = C_m(t) - q_- C_m(t), \text{ if } r(t) = 0, \text{ and pool } m \text{ choice} = \text{final choice}$$
$$C_m(t+1) = C_m(t) - q_d \times (C_m(t) - 0.5), \text{ if pool } m \text{ choice} \neq \text{final choice}$$
(11)

where $t$ represents the trial number, $C_m(t)$ is the synaptic strength of connections from object-value-encoding ($m = O$) or feature-value-encoding neurons ($m = F$), $q_d$ is the depression rate for the pool with a choice different than the final choice, and $q_+$ and $q_-$ are potentiation and depression rates, respectively.

As we have shown before, the decision only depends on the overall difference in the output of the two value-encoding pools[46–49]. This difference is proportional to the difference in the overall fraction of strong synapses in the two pools, since we assumed binary values for synaptic efficacy. Therefore, the probability of the final choice in the PDML model depends on the difference between the sum of the output of the value-encoding neurons selective for the presented objects or their features (shape and color):

$$\text{logit } P(O_1) = C_O(F_{O1} - F_{O2}) + C_F\left((F_{\text{shapeO1}} - F_{\text{shapeO2}}) + (F_{\text{colorO1}} - F_{\text{colorO2}})\right)/2\sigma$$
(12)

where $F_{\text{shape}Oi}(t)$ and $F_{\text{color}Oi}(t)$ are the synaptic strengths for synapses onto FVE neurons selective to shape and color, respectively. The probabilities of additional decisions (in DM circuits 1 and 2) based on the signal in an individual set of value-encoding neurons (OVE or FVE) are computed by setting $C_O$ or $C_F$ in the above equation to zero.

In the HDML model, a signal-selection circuit determines which set of the value-encoding neurons (OVE or FVE) contains a stronger signal first, and uses only the output of that set to drive the final DM circuit on a given trial. The probability of selecting the signal from OVE neurons, $P(\text{OVE})$, is computed using the following equation:

$$\text{logit } P(\text{OVE}) = C_O(F_{O1} - F_{O2}) - C_F\big((F_{\text{shape}O1} - F_{\text{shape}O2}) + (F_{\text{color}O1} - F_{\text{color}O2})\big)/2\sigma \tag{13}$$

Therefore, the final decision in the HDML model depends on the difference between the outputs of subpopulations in the set of value-encoding neurons which is selected as the set with stronger signal:

$$\text{logit } P(O_1) = (F_{O1} - F_{O2})/\sigma, \text{ if OVE signal is selected}$$
$$\text{logit } P(O_1) = \big(F_{\text{shape}O1} - F_{\text{shape}O2} + F_{\text{color}O1} - F_{\text{color}O2}\big)/2\sigma, \text{ if FVE signal is selected} \tag{14}$$

Finally, only plastic synapses from the value-encoding neurons with the stronger (hence chosen) signal to the signal-selection circuit are updated depending on the final choice and the reward outcome while the strength of the other set of plastic synapses decays to 0.5:

$$C_m(t+1) = C_m(t) + q_+(1 - C_m(t)), \text{ if } r(t) = 1$$
$$C_m(t+1) = C_m(t) - q_- C_m(t), \text{ if } r(t) = 0 \tag{15}$$
$$C_n(t+1) = C_n(t) - q_d \times (C_n(t) - 0.5)$$

where $m$ and $n$ denote the pools with the selected and unselected signals, respectively. It is worth noting that although the synaptic plasticity rule in our models relies on a single binary reward feedback to update reward values for different objects and features, the equivalent RL models based on reward prediction error (RPE) require separate RPE signals for updating different features.

**Models simulations**. In order to study the response of the PDML and HDML models to generalizability and frequency of changes in reward probabilities (volatility), we simulated each model over various environments (similar to those used in Experiments 1 and 2) with different levels of generalizability and volatility (Supplementary Fig. 9). More specifically, we linearly morphed a generalizable environment to a non-generalizable environment while modulating the level of volatility by changing the block length, $L$. To examine the interaction between dimensionality reduction and generalizability in adopting a model of the environment, we simulated various environments similar to those used in Experiments 3 and 4 (Supplementary Fig. 10). We changed the levels of generalizability by randomly shuffling some of the elements of the fully generalizable reward matrices with two values of dimensionality ($3^2$ and $4^2$). The reward probabilities were fixed over the course of these simulations, as in Experiments 3 and 4.

**Models parameters**. We used the following parameter values for all simulations of the PDML and HDML models presented in the paper, except otherwise mentioned (both models have six parameters): potentiation and depression rates for plastic synapses onto value-encoding neurons ($q_+ = q_- = 0.15$), potentiation and depression rates for plastic synapses onto the final DM circuit in the PDML model or signal-selection circuit in the HDML model ($q_+ = q_- = 0.075$), the depression rate for the rest of plastic synapses ($q_d = 0.015$), and the level of stochasticity in choice ($\sigma = 0.1$). Only for the simulations of the Experiment 4 presented in Fig. 7, the depression rate was set to a larger value ($q_d = 0.03$) in order to mimic greater forgetting of reward values in the experiment with a larger number of objects. Although we chose these specific parameter values for model simulations, the overall behavior of the models did not qualitatively depend on the exact values of these parameters.

**Assessment of models' response to different environments**. We assessed how the two models responded to properties of the environment, in terms of generalizability, volatility, and dimensionality, in three different ways. First, we measured performance, defined as the average harvested reward in a given environment. Second, we measured the difference in connection strengths from value-encoding neurons to the final DM circuit in the PDML model or to signal-selection circuit in the HDML model. The connection strengths from the OVE/FVE neurons to the final DM circuit in the PDML model or signal-selection circuit in the HDML model were equated with the synaptic strength ($C_O(t)$ and $C_F(t)$) in the respective models. Finally, we measured the difference in the overall weights that object-based and feature-based reward values exert on the final choice in each model.

In the PDML model, the strength of connections between each of the value-encoding neurons and the final DM circuit represents how strongly those neurons drive the final DM circuit. Similarly, the strength of connections between each of the value-encoding neurons and the signal-selection circuit represents how strongly those neurons drive the final DM circuit in the HDML model. In both models, however, the overall influence of the object-based or feature-based values on choice also depends on how signals encoded in plastic synapses onto the OVE and FVE neurons can differentiate between objects reward values. We computed such a 'differential signal' ($S$) for the object-based reward values by replacing $V_{O_i}(t)$ in Equation 3 with $F_{O_i}(t)$, which is the synaptic strength for synapses onto a pool $i$ of OVE neurons. Similarly, the differential signal for the feature-based reward values was computed by using the estimated reward values for objects based on the synaptic strengths for synapses onto FVE neurons selective to shape and color ($F_{\text{shape}, i}(t)$ and $F_{\text{color}, i}(t)$) and Equation 1.

Finally, the overall weight of the object-based and feature-based values on the final choice was computed using the product of the differential signal represented in a given set of value-encoding neurons and the strength of connections between those neurons and the final DM circuit in the PDML model or the signal-selection circuit in the HDML model. More specifically, the overall weight that the model assigned to the object-based reward value, $W_O(t)$, was set equal to $C_O(t) \times S_O(t)$ and the overall weight assigned to the feature-based reward value, $W_F(t)$, was set equal to $C_F(t) \times S_F(t)$.

**Code availability**. Computer codes that support the findings of this study are available from the corresponding author upon request.

**Data availability**. The data that support the findings of this study are available from the corresponding author upon request.

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

## Acknowledgements

We thank Deepak John and Suha Syed for help with earlier versions of the experiments. This work was supported by National Science Foundation (EPSCoR Award #1632738) grant to A.S., and National Institute of Health grants (MH108629 and MH108643) to D.L.

## Author contributions

A.S., D.L., K.R. and S.F. designed the study. K.R. and Z.A. performed the experiment. A.S., D.L., K.R. and S.F. analyzed the experimental data. A.S. and S.F. designed the model, preformed model simulations, and analyzed the data. A.S., D.L. and K.R. wrote the paper. All authors contributed to revising the manuscript.

## Additional information

**Competing interests:** The authors declare no competing financial interests.

