## [Peer Review File · Nature Communications]

Reviewers' comments:

Reviewer #1 (Remarks to the Author):

In this article, Farashahi and colleagues investigate a heuristic for learning that they propose could mitigate the curse of dimensionality in reinforcement learning. This heuristic proposes to learn the value of the multiple features that define an object, instead of learning the value of individual objects. The authors first show simulations highlighting why (and in which environments) this might be helpful for learning, in particular by generalizing information across objects. They then show results from four experiments supporting some model predictions. Last, they present a neural network implementation for arbitrating between the heuristic and object-based learning.

This is very interesting work, with well thought out theory and experiments, as well as promising results. However, I have a number of concerns that should be addressed to strengthen the conclusions that can be drawn from the paper. I summarize here my main issues, and detail them below. First, the authors rely very strongly on model-fitting for their conclusions. Model-fitting is a notoriously ambiguous tool, and should be supported with more detailed "raw" behavioral analysis, as well as various checks. My second issue is with the third part of the paper – relating to the implementation of the arbitration. This part, as is, seems very loosely related to the rest of the paper. The authors should attempt to unify the paper better. Last, a few conclusions are not well supported by the results. This should be changed.

1. Modeling vs. behavior

1a. As the authors state, the feature heuristic can approximate the true probabilities well – and of course, so can the objects-based RL. This begs the question of whether the two models are well identifiable in the four experiments. To test this, authors should simulate each model with a range of parameters, and fit the simulated behavior with each model, then show that each model fits better data simulated by the same model. This procedure also allows to verify that the parameters are identifiable, which is not a given in this kind of tasks.

1b. Model fitting results would be much more compelling if authors could show "raw" behavioral results that distinguish predictions from the two models. It doesn't seem particularly hard to identify such predictions in this case: for example, the feature based model (but not the object based model) predicts that reward to red triangle will make participants more likely to select a red circle/a blue triangle next trial, so plotting choice as a function of reward to a previous chosen object that is different but shares a feature would show a simple behavioral prediction in model simulations, and allow both testing of this effect in participants, and checking that models simulated with fit parameters do capture that effect (model validation step). This is only one example of how to support the model fitting findings with better behavioral analyses.

1c. p15, the authors discuss small learning rates as a marker of low attention to the corresponding feature. However, model-fitting with learning rates and softmax temperatures (here sigma) often lead to correlations between the two parameters (Daw 2011), such that it is difficult to interpret them individually (a low learning rate with a very low temperature could lead to better performance). Again, this result would be much more convincing if it was related to a raw behavioral result (for example same suggestion as 1b, but analyzed separately for the informative, and non informative features).

2. Supporting conclusions

2a. Fig. S3 seems to me to contradict a conclusion drawn p9: "we did not find any significant change in achieving this performance over the course of the experiment". Indeed, it appears that at the end of the experiment, participants are anticipating switches (performance dropping right before the end

of a schedule). Why did authors not choose to randomize length of the episodes?

2b. Fig. S4 also seems to contradict a conclusion at the top of p10 “indicating that the observed pattern of model adoption was not due to the use of different strategies early and late in the experiments”: indeed it seems that in the first two super blocks, participants do not favor either model, while they favor feature-based for the last two. This is an important aspect of the data in light of the questions that the authors ask in this paper, so should really be analyzed more carefully and explained.

2c. The last sentence of the first paragraph of p11 is also not supported – saying “led them to switch” supposes that participants go from feature based to object based RL, but there is no data to show this.

2d. Top of p16, regarding attentional bias to learning for informative features in high dimension – this is a strange conclusion to draw, since the result is present in dimension 3, but not in dimension 4.

2e. P12, the authors state a result regarding the participant-estimated probabilities but do not provide data or statistics to support this conclusion.

3. Integrating the last part of the article

The simulations shown in the last part for arbitrating between learners are interesting, but only barely related to the rest of the article. This is disappointing, since it could help understand the dynamical changes observed in behavior. I suggest the following. Authors could analyze the network model simulations as they did participants’ behavior (for example, fitting simulated network behavior with the object-based model and feature-based model). This could provide a way to qualitatively compare the networks’ behavior to the participants’ behavior, support the conclusion that the hierarchical model captures dynamics better, and thus to consolidate the paper.

Minor issues

A few concepts are talked about in the manuscript without being introduced first (they are introduced in the methods, but it should be possible to read the main text without jumping to methods). It would be good to either not talk about them, or quickly explain them. This includes the “coupled RL” models (p9), and the “bias” in the model (fig. 3).

Authors’ use of the term “model-based approach” in the introduction does not correspond to the way most current literature in RL interprets this term. Indeed, the literature uses “model-based” for RL behaviors that include planning in sequential tasks. While I agree with the authors that other meanings are possible, it would probably be confusing to most readers, thus I suggest not using this term here.

I recommend including Fig. S2 in the main text, rather than in the supplement – this would certainly help readers understand the paper better. I also don’t think table 1 is useful in the main text, as it’s redundant with figures. I would suggest moving it to the supplement.

One key aspect of the feature based RL model is that there is a different reward prediction error for each feature. This seems worth discussing, since the literature usually assumes that rpes are encoded in dopaminergic phasic signaling, and thus that there should only be one rpe.

The decay is implemented toward value 0 – is the initial starting point also 0? It is more adequate to

initialize and decay toward 0.5, as it provides equal weight to feedback of 0 vs. 1. When values are started at and decayed towards 0, it leads to lower effect of feedback 0, and bigger effect of feedback 1.

The title really doesn't reflect the content of the paper.

Reviewer #2 (Remarks to the Author):

The manuscript "Your favorite color makes learning more adaptable and precise" proposes a framework for learning in dynamic and complex environments based on either specific objects or abstract features of all available objects. Feature-based learning is thought to reduce dimensionality in complex environments but will be limited if the information carried by feature instances does not generalize. To test the proposed framework, the authors conduct four behavioural experiments that differ in complexity, volatility and generalizability. The experiments show that humans tend to use feature-based learning but switch to object-based learning in the absence of generalizability. Additionally, to provide a qualitative illustration of the findings, two neural networks (using hierarchical vs. parallel learning mechanisms) were set up and trained to learn based on object value or feature value. The hierarchical network provided a better overall characterization of the data than the parallel network.

The manuscript is well written and describes an interesting approach to assess learning, but I have some major concerns, particular with regard to the data handling and to the current description of the results. Please find my comments below.

Major concerns:

(1) Subject exclusion: For the second experiment, half of the subjects were excluded by the criteria of chance level choices. Also in the other experiments, several subjects were excluded who did not perform as expected. Is there any explanation why so many subjects couldn't make it above chance level (they were Dartmouth students after all)? Maybe they tried different strategies along the way (switching between features and objects)? These subjects should not only be excluded but analysed separately since the exclusion already fits the later tested models. Also, is it conceivable that individual differences may play a role such that some subjects by default learn in a more feature-based way (and these were excluded in experiments that could be solved only with object-based learning) and vice versa? By extension, one may then also wonder whether the environment plays a significant role over and above the proportion of one or the other type of learners in the sample.

(2) Overall, the results section of the manuscript lacks quantitative measures are missing at several points, such as:

- a. p. 9: "Both of these feature-based RLs provided better fits than their corresponding object-based RLs," – please quantify better fits (also in effect sizes)
- b. p. 9: "the coupled object-based RL and object-based RL with decay provided a significantly better fit than their corresponding feature-based RLs (Fig. 2c-e)." – please quantify significance and fits of whole Figure 2
- c. p. 14: "All of these results suggest stronger feature-based learning compared to object-based learning when dimensionality or generalizability increased, because both these quantities increased in Experiment 4 relative to Experiment 3" – how much is the increase and is it a significant difference?
- d. p. 20: "but the performance of the HDML model was slightly higher in all environments (Fig. 6a, d, g)." – please quantify 'slightly higher'
- e. p. 20: "HDML was more successful in assigning proper weights to different types of learning according to reward statistics in the environment" – what are proper weights?

f. p. 49: Timecourse of learning – can you provide any statistics, such as effect sizes (difference from chance) and correlations?

(3) Figure 3d and results: The split into trials 1-100 and 100-280 seems to be quite arbitrary and driven by the graphics for calculating a significant difference. Is there any (other) justification for this split and could a more principled split be used?

(4) Several subjects participated in more than one experiment: Was there any influence from previous participation? Did these subjects perform differently (better or worse)? Was within/between subject nature of comparisons accounted for in statistical analyses?

(5) Learning rate in experiments 3 and 4: Both learning rates (more inf. and less inf.) seem to be very small for these two experiments (even after exclusion of subjects) and it is unclear whether there really was meaningful learning at all.

a. Please report effect sizes in addition to p-values.

b. Please discuss these low learning rates critically discussed. How can we reliably say something about learning and differentiate between different models of learning if learning is so low?

c. Relatedly, is it not a matter of concern that the relative weight on bias is higher than the relative weight on object-based learning throughout the experiment (Fig. 3f)?

(6) Model comparison (Table 1) & Supplementary Figure 1: Are these results directly correlated? In other words, does in every experiment with a low generalizability the estimated error increase and therefore every model not based on the ability to abstract from specific objects (what is the definition here of feature-based) fit better? Please comment on this.

(7) Limitations of the new proposed framework are missing in the discussion.

(8) The model parameters (potentiation and depression rates, level of stochasticity, etc.) appear to be chosen arbitrarily. How plausible are they and can they be corroborated with independent research?

Minor points:

(1) The introduction should be carefully revised. The term of generalizability should be introduced and defined here, since it is a central part of the manuscript (and generalizability index should be defined in the methods section). Also, the description of available frameworks is rather short and doesn't show the need for a new framework. Additionally, the last paragraph mentions hierarchical decision-making out of the blue.

(2) What are real-world estimates of generalizability and dimensionality?

(3) What is the learning rate used in Figure 1b?

(4) P. 9: "coupled" comes out of the blue

(5) P. 12/Supplementary Fig. 5a: it is unclear in which sense one feature was partially informative. Please provide example/more detail.

(6) Fig. 6 h, I (maybe g): Plots are meant to show HDML-PDML (which would be desirable) but currently seem to show PDML-HDML.

(7) On p. 24, the reference to Kahnt, Chang, Park, et al. 2012 (Journal of Neuroscience) should be replaced with Kahnt, Park, Burke, et al. 2012 (Journal of Neuroscience) or Kahnt & Tobler, 2016

(Elife).

(8) P. 25/26: There may be a perceived discrepancy between the potential need for arbitration stated on p. 25 and the statement that “reward feedback alone can correctly adjust behavior toward a more object-based or amore feature-based approach, without any explicit optimization”. Please clarify

(9) On p. 26: “The HDML model proposed in this study shares some components with the model of Hunt et al. (2014), though our model includes learning as well. Similarly, Wunderlich et al (2011) also suggested that the brain holds weights for all possible informative dimensions simultaneously, and these weights are updated on every trial.” – Please describe shared components with a bit more specificity.

(10) What software has been used for the computer experiments?

(11) What was the age range of the subjects?

(12) Which Matlab version has been used for analysis?

(13) Supplementary Figure 2: Superblock length was 196 trials, made up of four blocks of 48 =192 trials. What were the remaining four trials?

(14) Supplementary Figure 3: Given that the maximum performance was considerably higher in the generalizable compared to the non-generalizable experiment, it is unclear why the average performance does not differ/how maximum was defined (in any case, the maximum definition should be described as it used also elsewhere).

(15) Supplementary Figure 3: Please replace “strategies” with “performance” in the last sentence of the figure legend (there could have been different strategies but if so they did not translate into differences in performance).

Some typos

(1) P. 5: “how the performance of the two learners depend” -> “depends”

(2) P. 5: “based of features” -> “based on features”

(3) P. 16: add “the” between “any of” and “four experiments”

(4) P. 25: delete “and” between “circuits,” and “arbitration”

(5) P. 49: Legend of Supplementary Figure 3 should refer to Supplementary Figure 2, not Supplementary Figure 1.

Reviewer #3 (Remarks to the Author):

The study by Farashahi and colleagues presents behavioural and modelling results addressing the question of how we learn in multidimensional environments – specifically, how we deal with the “curse of dimensionality”. This study addresses a key topic in computational neuroscience and the authors present cleverly designed behavioural experiments combined with elegant modelling – both algorithmic and neural network models to address this issue.

Learning the values of individual objects is most precise, but may require many trials when there are several combinations of features - specifically, the number of possible objects is n to the power of m , where m is the number of features (e.g. colour) and n is the number of values for each feature (e.g. blue, red, green). Thus there is a trade-off between accuracy and the time it takes to learn, which becomes a particular issue when environments are non-stable.

Using simulations, the authors first show that an object-based learner outperforms a feature-based learner later during learning, and this is further shifted as a function of generalizability of the environment (i.e. the better actual object reward probabilities can be inferred from feature-based values). Thus, the more generalizable the environment, the more time it takes for an object-based learner to outperform a feature-based learner.

They then present four behavioural experiments. In experiments 1 and 2, the environment was non-stationary. It was found that subjects adopted feature-based learning even though this did not reduce the dimensionality, but only when the environment was generalizable (experiment 1). In contrast, when the environment was non-stationary but did not allow generalisation, subjects adopted an object-based strategy. In experiments 3 and 4, the environment was kept stationary, while increasing the dimensionality of the state space. This allowed them to isolate the influence of dimensionality reduction. Both in experiment 3 and 4, subjects initially adopted feature-based learning and then shifted to an object-based strategy later on. However, in experiment 4, the increased dimensionality favoured the feature-based strategy – model comparison showed that behaviour was better explained by an object-based learner in experiment 3, whereas the feature-based learner provided a better fit in experiment 4.

Next, the authors set up two neural network models to provide a mechanistic account for the observed learning behaviour, PDML and HDML. In both models, there were two sets of value-coding neurons learn reward values of individual objects (OVE) or features (FVE). Synapses onto both sets of neurons undergo reward-dependent plasticity. The two models differ in terms of how the output of OVE and FVE is used to generate choice. In the PDML, both OVE and FVE feed into a common (final) decision circuit. There is however also a decision circuit that generates a choice based on the output of OVE and FVE, respectively. If a choice results in reward, then the connections between the set that produced the same choice (OVE or FVE) and the final decision circuit are strengthened. By contrast, in the HDML, there is a single selection circuit in which the outputs of OVE and FVE compete to control which element ultimately governs the decision. Performance of the HDML was somewhat better than that of PDML in all four environments. Likewise, the HDML adapted better to differences in volatility or generalizability of the environment.

This is a very elegant study that mechanistically dissects human reinforcement learning. I only have a few points that I would like the authors to clarify.

- 1) The network modelling results are elegant and show that the HDML and PDML perform the task reasonably well and that the HDML adapts to the key parameters of the environment better than does the PDML. What I am somewhat missing is how the behaviour of the model compares to real subjects. Even if the behaviour is qualitatively similar, it would be good to have a more direct comparison of network model behaviour with experimental data.

- 2) The algorithmic models were fit to subject data using Nelder-Mead simplex algorithm implemented in Matlab's `fminsearch`. This appears to be a suboptimal choice to me – this algorithm is notorious for getting stuck in local minima – there are other ways that would certainly be better. Given the low

number of parameters in the models, I would suggest at least using a grid search to find an educated starting point for the simplex. Since the models are integral to this manuscript, it is important that the modelling results are immune to any such criticism.

3) My understanding is that both in the PDML and HDML, the synapses from both OVE and FVE to the decision circuit undergo reward-dependent plasticity. If this is true, this is not clearly evident from the text and figure 5, where it appears that only the synapses onto OVE/FVE neurons are subject to such plasticity. I would ask the authors to clarify this point.

4) The experiments are designed in a very clever and sophisticated manner – which however sometimes makes it difficult for a reader to precisely understand what exactly was done. In particular, I found it hard to grasp from supplementary figure 2 how the different manipulations in terms of dimensionality/generalizability were implemented. In fact, I felt more confused after looking at the figure than before. Can you please think of a way of making this figure and the description easier to follow? Specifically, what is shown in part B – is this different schedules in each line? But then why is there a different symbol to the left on each side? This suggests I should read the plot column-wise – please help this with a clearer description.

5) I think there is an important recent paper from Yael Niv's lab (Leong et al., 2017, Neuron) that also addresses the curse of dimensionality and the propose a two-way interaction between learning and attention. While their contribution is clearly distinct from the approach presented here, I think this study should be given credit here as there are direct links.

Response to the reviewers' comments, and summary of changes made in response to the comments of the reviewers.

Title: Your favorite color makes learning more adaptable and precise

Authors: Farashahi, Rowe, Aslami, Lee, and Soltani

We are greatly thankful to all the reviewers for their careful reading of our manuscript and for their useful and constructive suggestions. We have performed additional analyses and made substantial changes in the revised manuscript to fully address all of the reviewers' concerns and suggestions. Below, we provide point-by-point responses to individual reviewers' comments/concerns. The corresponding changes have been clearly marked and noted in the revised manuscript (e.g. [R1.1], etc.).

Reviewer #1

"In this article, Farashahi and colleagues investigate a heuristic for learning that they propose could mitigate the curse of dimensionality in reinforcement learning. This heuristic proposes to learn the value of the multiple features that define an object, instead of learning the value of individual objects. The authors first show simulations highlighting why (and in which environments) this might be helpful for learning, in particular by generalizing information across objects. They then show results from four experiments supporting some model predictions. Last, they present a neural network implementation for arbitrating between the heuristic and object-based learning. This is very interesting work, with well thought out theory and experiments, as well as promising results. However, I have a number of concerns that should be addressed to strengthen the conclusions that can be drawn from the paper. I summarize here my main issues, and detail them below. First, the authors rely very strongly on model-fitting for their conclusions. Model-fitting is a notoriously ambiguous tool, and should be supported with more detailed "raw" behavioral analysis, as well as various checks. My second issue is with the third part of the paper – relating to the implementation of the arbitration. This part, as is, seems very loosely related to the rest of the paper. The authors should attempt to unify the paper better. Last, a few conclusions are not well supported by the results. This should be changed."

Response: We thank the reviewer for a positive evaluation of our work. We hope that our answers here and the corresponding changes in the revised manuscript addressed all the reviewer's concerns.

“1. Modeling vs. behavior

1a. As the authors state, the feature heuristic can approximate the true probabilities well – and of course, so can the objects-based RL. This begs the question of whether the two models are well identifiable in the four experiments. To test this, authors should simulate each model with a range of parameters, and fit the simulated behavior with each model, then show that each model fits better data simulated by the same model. This procedure also allows to verify that the parameters are identifiable, which is not a given in this kind of tasks.”

Response: We thank the reviewer for pointing out this issue about the ability to identify the correct model based on fitting choice data. As suggested by the reviewer, to address this issue we have simulated choice data in all four experiments using the same six models utilized for fitting (three are based on object-based learning and three are based on feature-based learning) over a wide range of model parameters. We then fit the simulated data with all these models to compute the goodness-of-fit and to estimate original model parameters.

We found that for most models and experiments the model used to generate the data provided the best fit for generated data, with 4 exceptions (out of 24; see Figure 1 below). Even for those cases (coupled feature-based or object-based models during Experiments 3 and 4), the fits based on the models with similar learning type (object-based or feature-based) were better than those with different learning types, indicating that the object-based or feature-based nature of learning is identifiable in all cases.

Moreover, we obtained the minimum value of the average difference between the actual and estimated model parameters for the same model used for generating the data, with a few exemptions similar to what was found for fitting (see Figure 2 below). Note that for this analysis we used the ratio of the learning rate to the stochasticity in choice because these two parameters influence choice similarly (i.e. a scaled version of the two parameters results in very similar choice behavior) and are correlated as pointed out by the reviewer (see [R1.3] below). Together, these analyses revealed that our fitting method can be used to distinguish between the alternative models and to estimate model parameters accurately.

We have discussed these points in the revised manuscript, and added these results as Supplementary Figures 4 and 5 (see [R1.1]).

Figure 1. Comparison of the goodness-of-fit for the simulated data using various models in Experiments 1 to 4. Each column shows results generated with a given model (numbered 1 to 6) and row **a** to **d** correspond to Experiments 1 to 4, respectively. Plotted is the average negative log likelihood ($-LL$) over all sets of parameters (mean \pm s.e.m.) for data generated with one of the six models in Experiments 1 to 4 and fit with each of the six models. The results for the model used to generate data in a given experiment and its object-based or feature-based counterpart are highlighted in cyan and orange, respectively.

Figure 2. Error in estimated model parameters for the simulated data using various models. Each column shows results generated with a given model (numbered 1 to 6) and row **a** to **d** correspond to Experiments 1 to 4, respectively. Plotted is average absolute error in estimating the ratio of the learning rate to the stochasticity in choice over all sets of parameters (mean \pm s.e.m.) for data

generated with one of the six models in Experiments 1 to 4 and fit with each of the six models. The results for the model used to generate data in a given experiment and its object-based or feature-based counterpart are highlighted in cyan and orange, respectively.

“1b. Model fitting results would be much more compelling if authors could show “raw” behavioral results that distinguish predictions from the two models. It doesn’t seem particularly hard to identify such predictions in this case: for example, the feature based model (but not the object based model) predicts that reward to red triangle will make participants more likely to select a red circle/a blue triangle next trial, so plotting choice as a function of reward to a previous chosen object that is different but shares a feature would show a simple behavioral prediction in model simulations, and allow both testing of this effect in participants, and checking that models simulated with fit parameters do capture that effect (model validation step). This is only one example of how to support the model fitting findings with better behavioral analyses.”

Response: We thank the reviewer for mentioning this point and for their useful suggestion in testing the predictions of feature-based learning using raw behavioral data.

We performed the suggested analysis by computing the feature-based ‘differential response’ that measures the change in the value of features of the object selected on the previous trial. More specifically, we defined the feature-based differential response equal to the probability of selecting an object that contains only one of the two features of the object selected and rewarded on the previous trial (i.e. rewarded object) when this object was paired with an object that did not share any feature with the previously rewarded object. For example, denoting the probability of choosing X when it is presented together with Y in trial t given that choosing Z was rewarded and unrewarded in the previous trial as $p(X \succ Y(t) | Z(t-1)+)$ and $p(X \succ Y(t) | Z(t-1)-)$, $p(\square \succ \triangle(t) | \blacksquare(t-1)+) - p(\square \succ \triangle(t) | \blacksquare(t-1)-)$ measures the change in the value of features of the object selected on the previous trial. For comparison, we also calculated the object-based differential response equal to the difference between the probability of selecting the object that was selected and rewarded on the previous trial and the same probability when the previous trial was not rewarded (e.g. $p(\text{choose } \blacksquare \text{ in trial } t | \blacksquare(t-1)+) - p(\text{choose } \blacksquare \text{ in trial } t | \blacksquare(t-1)-)$). The object-based differential response measures the change in preference for an object due to reward feedback and is equivalent to the difference between win-stay and lose-switch strategy for a particular object. We found that, indeed, subjects who adopted feature-based learning updated their preference for objects that shared a feature with the object selected on the previous trial depending on the reward outcome (blue histograms in Figure 3a-d below).

This was not the case for subjects who adopted object-based learning (blue histograms in Figure 3e-h below).

Moreover, as suggested by the reviewer, we validated the model used for fitting by simulating choice data based on the estimated model parameters for individual subjects and computing the feature-based and object-based differential responses for the simulated data. The similarity between the average differential responses based on simulated and actual choice (dashed and solid lines in Figure 3 below) indicated the success of our models in capturing behavior.

The results of this analysis have been added to the revised manuscript as a new section (Testing the behavioral predictions of feature-based learning). We have also added a new Figure 4 and revised the manuscript to discuss relevant Methods and results (see [R1.2]).

Figure 3. Subjects who adopted feature-based learning updated their preference even for objects that contained a feature of the object selected on the previous trial based on the reward outcome. (a-d) Plotted are the feature-based (blue) and object-based (red) differential responses for subjects who adopted feature-based learning in a given experiment. The dashed lines show the median values across subjects a star indicates significant difference from zero (one-sided sign-rank test, $P < 0.05$). The solid lines show the average simulated differential response using the estimated parameters based on the fit of each subject's data. (e-h) The same as in a-d but for subjects who adopted object-based learning in each experiment.

“1c. p15, the authors discuss small learning rates as a marker of low attention to the corresponding feature. However, model-fitting with learning rates and softmax temperatures (here sigma) often

lead to correlations between the two parameters (Daw 2011), such that it is difficult to interpret them individually (a low learning rate with a very low temperature could lead to better performance). Again, this result would be much more convincing if it was related to a raw behavioral result (for example same suggestion as 1b, but analyzed separately for the informative, and non informative features).”

Response: We thank the reviewer for raising the concern that the correlation between the estimated learning rates and the softmax temperature (which we refer to as the stochasticity in choice) could contaminate the analyses presented in previous Figure 4.

Indeed, we found a significant positive correlation between the two parameters within individuals in all experiments (see Figure 4 below). Therefore, we revised the analysis so that rather than comparing between the estimated learning rates and weights of the more and less informative features, it compared the product of these parameters. This product can measure the overall amount of change in choice behavior due to reward feedback. This analysis revealed larger values for the product of the learning rate and the assigned weight for the more informative feature in all experiments, indicating that subjects incorporated reward feedback (in terms of a combination of the learning rate and weights) from this feature more strongly (see Figure 5a-d below).

As suggested by the reviewer, we also computed the feature-based differential response for the more and less informative features (see Figure 5e-h below) and found this quantity to be larger for the more informative feature in all experiments (though it did not achieve significance in Experiments 2 and 3 due to the small sample size), indicating that subjects updated their behavior more strongly for the more informative feature.

Figure 4. Correlation between the stochasticity in choice and the estimated learning rates for individual subjects during the four experiments.

Figure 5. Feature-based learning was stronger for the more informative feature. (a-d) Plotted is the log product of the estimated learning rate (α) and assigned weight (w) for the less informative feature (non-informative in the case of Experiments 2-4) versus of the same product for the more informative feature for each individual, across four experiments. The insets show the histogram of difference in ($\alpha * w$) between the more and less informative features. The dashed lines show the medians and the solid gray lines indicate zero. The star shows that the median of the difference in ($\alpha * w$) were significantly different from 0 (one-sided sign-rank test, $P < .05$). These products were larger for the more informative feature in all experiments. (e-h). Plotted is the feature-based differential response for the less informative feature versus the more informative feature. Conventions are the same as in panels a-d. The feature-based differential response was larger for the more informative feature in all experiments (though it did not achieve significance in Experiments 2 and 3), indicating that subjects updated their behavior more strongly for the more informative feature.

The results of these analyses have been summarized in the new Figure 5 and the corresponding texts have been revised to include these results (see [R1.3]).

“2. Supporting conclusions

2a. Fig. S3 seems to me to contradict a conclusion drawn p9: “we did not find any significant change in achieving this performance over the course of the experiment”. Indeed, it appears that at the end of the experiment, participants are anticipating switches (performance dropping right before the end of a schedule). Why did authors not choose to randomize length of the episodes? “

Response: We thank the reviewer for pointing out this possible issue. We did not randomize the length of the blocks and instead used equal number of trials in order to control the probability of reward assignment on the four objects in a given block.

Moreover, we did not expect that subjects could possibly track reversals over such a long block of trials ($L = 48$).

To test whether the reviewer’s observation is an artifact of the filtering, noise, or a real anticipation of reversals, we computed the average performance over every super-block of trials (four blocks made a super-block of 192 trials) during Experiments 1 and 2. We also computed the probability of choosing the better option on a given trial throughout the course of the two experiments. In contrast to the performance, which reflects factors other than subject’s preference such as the actual values of reward probability, the probability of selecting the better option captures how subjects discriminated between the four options based on their estimates of reward value, and thus could reveal any switch in choice behavior independently of the actual values of reward probability.

These analyses revealed that subjects gradually learned about the reward value of the four options after each reversal, and this learning was similar between the early and late super-blocks of the experiments (see Figure 6 below). However, they also showed a decrease in learning during the last super-block of Experiment 2, indicated by lower probability of selecting the more rewarding option in the last super-block. Together, we did not find any evidence for anticipation of reversals and thus, believe that the referred drop in performance is simply caused by noise amplified by filtering with a boxcar (which had the length of 10 in the previous Supplementary Figure 3).

Figure 6. Subjects’ performance and choice behavior over the course of Experiments 1 and 2. (a-d) Plotted is average performance during the four super-blocks of Experiments 1 and 2. The shaded areas indicate s.e.m. The dashed line shows the chance performance. (e-h) Plotted is the average probability of choosing the more rewarding option on each trial during the four super-blocks.

In the revised manuscript, we have replaced the previous Supplementary Figure 3 with the Figure 5 above to avoid future confusion, revised the text accordingly, and have mentioned the drop in performance in the last super-block of Experiment 2 (see [R1.4]).

“2b. Fig. S4 also seems to contradict a conclusion at the top of p10 “indicating that the observed pattern of model adoption was not due to the use of different strategies early and late in the experiments”: indeed it seems that in the first two super blocks, participants do not favor either model, while they favor feature-based for the last two. This is an important aspect of the data in light of the questions that the authors ask in this paper, so should really be analyzed more carefully and explained.”

Response: We thank the reviewer for pointing this potential issue in the interpretation of our results. It seems that the reviewer is referring to the median of the difference in the goodness-of-fit (in terms of BIC) not being significantly smaller than 0 in all super-blocks of Experiment 1. We do not believe that such lack of significance in some super-blocks reported in the previous Supplementary Figure 4 constitutes a change in strategy, because the differences in goodness-of-fit based on the two models during all superblocks were consistent with those based on fits of the entire experiment. Indeed, after updating the models with the decay based on another reviewer’s suggestion (see [R1.11] below) the pattern of goodness-of-fit slightly changed (see Figure 7 below) further indicating that the lack of significance does not necessarily mean a change in learning strategy.

Figure 7. Comparison of the goodness-of-fit based on the best object-based and feature-based models for each individual during each super-block of Experiments 1 and 2. The insets show histograms of the difference in the BIC from the two best models for the generalizable (blue) and non-generalizable (red) environments. The dashed lines show the medians, and the star shows that the median is significantly different from zero (one-sided sign-rank test, $P < 0.05$).

We have updated the above figure (now part of Supplementary Figure 6) and discussed its results more cautiously in the revised manuscript (see [R1.5]).

“2c. The last sentence of the first paragraph of p11 is also not supported – saying “led them to switch” supposes that participants go from feature based to object based RL, but there is no data to show this. ”

Response: In the aforementioned paragraph, we were referring to better quality of fit for feature-based models in Experiment 1 (generalizable environment) and better quality of fit for object-based models in Experiment 2 (non-generalizable environment). That is, we referred to ‘groups’ of subjects changing their behavior between Experiments 1 and 2 and not switch between models for a given individual, which seems to cause the confusion.

This has been clarified in the revised manuscript (see [R1.6]).

“2d. Top of p16, regarding attentional bias to learning for informative features in high dimension – this is a strange conclusion to draw, since the result is present in dimension 3, but not in dimension 4.”

Response: We thank the reviewer for pointing out this inaccuracy. These results presented in that section are updated based on the new analyses suggested by Reviewers # 1 and 2. We now can observe significant effects of bias for learning from reward feedback for the more informative feature except for two sets of analyses in Experiments 2 and 3 due to the small sample size (see Figure 5 above).

The entire section on the influence of attention on feature-based learning has been revised accordingly (see [R1.7]).

“2e. P12, the authors state a result regarding the participant-estimated probabilities but do not provide data or statistics to support this conclusion.”

Response: The reviewer seems to refer to the results presented in Figure 3b-c. For the first set of results about the relative weight, we provided the confidence intervals to show that the difference is significant (this information appeared on different pages which could result in missing the statistics). For the second set of results based on the correlation analysis (Fig. 3c), we now have added chi-square test to show that the proportion of subjects that show more correlation with actual reward probabilities indeed increased over time.

The new statistical tests have been added to the revised manuscript (see [R1.8]).

“3. Integrating the last part of the article

The simulations shown in the last part for arbitrating between learners are interesting, but only barely related to the rest of the article. This is disappointing, since it could help understand the dynamical changes observed in behavior. I suggest the following. Authors could analyze the network model simulations as they did participants’ behavior (for example, fitting simulated network behavior with the object-based model and feature-based model). This could provide a way to qualitatively compare the networks’ behavior to the participants’ behavior, support the conclusion that the hierarchical model captures dynamics better, and thus to consolidate the paper.”

Response: We thank the reviewer for pointing out the disconnection between the modeling and experimental parts of the paper, and for the helpful suggestions for resolving this issue.

As suggested by the reviewer, we simulated choice behavior of the PDML and HDML models during all four experiments and fit the simulated choice data based on the same models used to fit the experimental data (see Figure 8 below). We found that for Experiment 1, the simulated data using the PDML model were equally fit by the feature-based and object-based models (Fig. 7a) indicating that PDML could not adopt feature-based learning in a volatile, generalizable environment. In contrast, choice behavior of the PDML model during Experiment 2 was better fit by the object-based model similarly to the experimental data (Figure 8a below). Choice behavior of the HDML model was consistent with our results in both Experiments 1 and 2 (Figure 8f below). For Experiment 3, both models adopted feature-based learning first and slowly transitioned to object-based learning (Figure 8b, g below) and moreover, their choice behavior was better fit by object-based learning later in the experiment (Figure 8c, h below). Both models also adopted feature-based learning first during Experiment 4 but showed only a small transition toward object-based learning (Figure 8d, i below) such that their choice behavior was still better fit by feature-based learning even toward the end of the experiment (Figure 7e, j below). These patterns were consistent with those of experimental data during Experiments 3 and 4. Overall, choice behavior of the HDML model qualitatively matched the pattern of data in all experiments whereas the PDML model failed to capture choice behavior during Experiment 1.

The below Figure 8 has been added as a new Figure 7 to the revised manuscript, and a new paragraph has been included to discuss these results (see [R1.9]).

Figure 8. Replicating the pattern of experimental data using the PDML and HDML models. (a) Comparison of the goodness-of-fit in for the data generated by the PDML model in Experiments 1 (generalizable) and 2 (non-generalizable) using the object-based and feature-based RL models with decays. The insets show histograms of the difference in the negative log likelihood based on the fits of the two models. (b) The time course of model adoption in the PDML model. Plotted is the relative weight of object-based to the sum of the object-based and feature-based weights, and explained variance in estimates (R^2) over time in Experiment 3. Dotted lines show the fit of data based on an exponential function. (c) Transition from feature-based to object-based learning in the PDML model. Plotted are the average negative log likelihood based on the best feature-based model, best object-based RL model, and the difference between object-based and feature-based models in Experiment 3. Shaded areas indicate s.e.m., and the dashed line shows the measure for chance prediction. (d-e) The same as in b-c, but for simulations of Experiment 4. (f-j) The same as in a-e, but for the HDML model. Overall, choice behavior of the HDML model was more consistent with experimental data.

Minor issues

“A few concepts are talked about in the manuscript without being introduced first (they are introduced in the methods, but it should be possible to read the main text without jumping to methods). It would be good to either not talk about them, or quickly explain them. This includes the “coupled RL” models (p9), and the “bias” in the model (fig. 3). “

Response: We thank the reviewer for pointing out this issue. We have made changes throughout the revised manuscript to fix this and similar issues.

“Authors’ use of the term “model-based approach” in the introduction does not correspond to the way most current literature in RL interprets this term. Indeed, the literature uses “model-based” for RL behaviors that include planning in sequential tasks. While I agree with the authors that other

meanings are possible, it would probably be confusing to most readers, thus I suggest not using this term here."

Response: We thank the reviewer for pointing out this issue. We have removed "model-based approach" in the Introduction.

"I recommend including Fig. S2 in the main text, rather than in the supplement – this would certainly help readers understand the paper better. I also don't think table 1 is useful in the main text, as it's redundant with figures. I would suggest moving it to the supplement. "

Response: We thank the reviewer for pointing out this issue. We wish we could move that supplementary figure to the main text, but we are afraid that is not feasible after the addition of three new figures to the main text based on new analyses requested by the reviewers. We have moved Table 1 to the supplement.

"One key aspect of the feature based RL model is that there is a different reward prediction error for each feature. This seems worth discussing, since the literature usually assumes that rpes are encoded in dopaminergic phasic signaling, and thus that there should only be one rpe."

Response: We thank the reviewer for pointing this out.

We now have noted this connection in the Discussion (see [R1.10])

"The decay is implemented toward value 0 – is the initial starting point also 0? It is more adequate to initialize and decay toward 0.5, as it provides equal weight to feedback of 0 vs. 1. When values are started at and decayed towards 0, it leads to lower effect of feedback 0, and bigger effect of feedback 1."

Response: We thank the reviewer for pointing this out. We adopted decay toward 0 based on the original model by Niv et al. (2015). However, as mentioned by the reviewer, it makes more sense (especially in the context of binary choice) for the value to decay back to 0.5 (i.e. the state with no information) when options are not selected.

In the revised manuscript, we updated both our model fitting procedures and model simulations so that reward values are initialized from 0.5 and decay to 0.5 if corresponding options are not selected (see [R1.11]).

“The title really doesn’t reflect the content of the paper. “

Response: We respectfully submit that the title captures the main point of the paper in the sense that choosing a favorite color (i.e. adopting feature-based learning) can partially mitigate the adaptability-precision tradeoff. Nevertheless, we would still welcome any suggestions from reviewers and editors.

Reviewer #2:

“The manuscript “Your favorite color makes learning more adaptable and precise” proposes a framework for learning in dynamic and complex environments based on either specific objects or abstract features of all available objects. Feature-based learning is thought to reduce dimensionality in complex environments but will be limited if the information carried by feature instances does not generalize. To test the proposed framework, the authors conduct four behavioural experiments that differ in complexity, volatility and generalizability. The experiments show that humans tend to use feature-based learning but switch to object-based learning in the absence of generalizability. Additionally, to provide a qualitative illustration of the findings, two neural networks (using hierarchical vs. parallel learning mechanisms) were set up and trained to learn based on object value or feature value. The hierarchical network provided a better overall characterization of the data than the parallel network.

The manuscript is well written and describes an interesting approach to assess learning, but I have some major concerns, particular with regard to the data handling and to the current description of the results. Please find my comments below.”

Response: We thank the reviewer for a positive evaluation of our work. We hope that our new analyses, answers provided here, and corresponding changes in the revised manuscript addressed all the reviewer’s concerns.

“Major concerns:

(1) Subject exclusion: For the second experiment, half of the subjects were excluded by the criteria of chance level choices. Also in the other experiments, several subjects were excluded who did not perform as expected. Is there any explanation why so many subjects couldn’t make it above chance level (they were Dartmouth students after all)? Maybe they tried different strategies along the way (switching between features and objects)? These subjects should not only be excluded but analysed

separately since the exclusion already fits the later tested models. Also, is it conceivable that individual differences may play a role such that some subjects by default learn in a more feature-based way (and these were excluded in experiments that could be solved only with object-based learning) and vice versa? By extension, one may then also wonder whether the environment plays a significant role over and above the proportion of one or the other type of learners in the sample.

Response: We thank the reviewer for pointing out the possible issue with excluding certain subjects. As mentioned by the reviewer, we excluded subjects based on the same chance criterion in each experiment. This resulted in the exclusion of about half of subjects in Experiment 2 because this experiment was the most difficult experiment as it involved an environment that was both non-generalizable and volatile (this difficulty was reflected in an overall lower performance in this experiment). Therefore, task difficulty could cause more subjects to forgo engaging and learning in all experiments in general and in Experiment 2 in particular.

To address the reviewer's concern that the exclusion criterion could bias the interpretation of experimental results in certain ways, we also analyzed data from all excluded subjects in all four experiments (as suggested by the reviewer). This analysis, which is summarized in Figures 9 and 10 below, did not reveal that excluded subjects adopted a strategy qualitatively different from the one used by the remaining subjects in a given experiment, and their pattern of choice resembled either feature-based or object-based learning due to chance. In other words, excluded subjects did not favor feature-based or object-based learning and instead failed to learn reward values in a given experiment. We could identify 3 subjects who adopted very specific strategies during Experiment 2 to avoid learning (and re-learning) all four options, resulting in their poor performance and thus exclusion. One subject only selected the option on the right side of the screen throughout the experiment, and other two selected left or right options ~80% of the time (those with small BIC values in Figure 9c below). We were not able to identify any other specific strategy for other excluded subjects.

These results have been discussed in the revised manuscript (see [R2.1]) and the Figures 9 and 10 above has been added as the new Supplementary Figure 7 and a part of the new Supplementary Figure 6, respectively.

Figure 9. Analyses of choice behavior and estimation of the excluded subjects. (a-b) Time course of learning during each block of trials in Experiments 1 and 2. Plotted is the average harvested reward (a) or the probability of selecting the better option on a given trial within a block across all excluded subjects. The dashed line shows chance performance. The solid blue and red lines in panel a show the maximum performance based on the feature-based approach in the generalizable and non-generalizable environments, respectively. The maximum performance for the object-based approach was similar in the two environments, and equal to that of the feature-based approach in the generalizable environment [R2.14]. Overall, these subjects failed to learn reward probabilities associated with four options during most of each block. (c) Plotted is the Bayesian information criterion (BIC) based on the best feature-based or object-based models for a given subject, separately for each environment. The insets show histograms of the difference in BIC from the feature-based and object-based models, and the dashed lines show the medians, which is on top of the zero line. There was no evidence that the excluded subjects adopted a strategy qualitatively different from the one used by other subjects. (d) The time course of performance during Experiment 3. Shaded areas indicate s.e.m., and the dashed line shows chance performance. The red and blue solid lines show the maximum performance using the feature-based and object-based approaches, respectively, assuming that the decision maker selects the more rewarding option based on a given approach on every trial. Arrows mark the locations of estimation blocks throughout a session. (e) The time course of model adoption measured by fitting subjects'

estimates of reward probabilities. Plotted is the relative weight of object-based to the sum of the object-based and feature-based approaches, and explained variance in estimates (R^2) over time. Dotted lines show the fit of data based on an exponential function. (f) Transition from feature-based to object-based learning revealed by the average goodness-of-fit over time. Plotted are the average negative log likelihood based on the best feature-based model, best object-based RL model, and the difference between the best object-based and feature-based models during Experiment 3. Shaded areas indicate s.e.m., and the dashed line shows the measure for chance prediction. Overall, excluded subjects moved toward object-based learning over time similarly to what found for the subjects included in the study. (g-i) The same as in d-f, but during Experiment 4. Throughout this experiment, feature-based learning provided a better fit for choice behavior of excluded subjects similarly to what was found for the subjects included in the study.

Figure 10. Comparison of the goodness-of-fit based on the best object-based and feature-based models for each individual (excluded subjects) during each super-block of Experiments 1 and 2.

Finally, we agree with the reviewer that individual differences could play a role, and the environment in the real world could push some subjects toward one of the two types of learning strategy. However, there is no a priori evidence that such effects should cause subjects to adopt a specific model for a given experiment. Moreover, random assignment of subjects to different experimental conditions can mitigate the issue of individual variability. We believe that addressing how much of the observed individual variability is due to innate tendency compared to the experimental condition or chance is beyond the scope of our study.

“(2) Overall, the results section of the manuscript lacks quantitative measures are missing at several points, such as:

a. p. 9: “Both of these feature-based RLs provided better fits than their corresponding object-based RLs, ...” – please quantify better fits (also in effect sizes)

b. p. 9: “the coupled object-based RL and object-based RL with decay provided a significantly better fit than their corresponding feature-based RLs (Fig. 2c-e).” – please quantify significance and fits of

whole Figure 2

c. p. 14: “All of these results suggest stronger feature-based learning compared to object-based learning when dimensionality or generalizability increased, because both these quantities increased in Experiment 4 relative to Experiment 3” – how much is the increase and is it a significant difference?

d. p. 20: “but the performance of the HDML model was slightly higher in all environments (Fig. 6a, d, g).” – please quantify ‘slightly higher’

e. p. 20: “HDML was more successful in assigning proper weights to different types of learning according to reward statistics in the environment” – what are proper weights?

f. p. 49: Time course of learning – can you provide any statistics, such as effect sizes (difference from chance) and correlations?”

Response: We thank the reviewer for pointing out places where we did not report the results of statistical tests. In some places, it was a mistake and in some places the statistical tests were not included to avoid repetition with the figure captions.

To solve both issues, we now have provided statistical tests for aforementioned places (except “c” and “e”) and have removed the corresponding texts in the figure captions. For “c”, the dimensionality and generalizability in the two environments are specific values and now are reported (previously reported in Figure 7 and Methods). For “e”, we have revised the sentence to be more accurate. For “f”, the shaded area in the Supplementary Figure 3 indicated the s.e.m. and thus showed a significant difference from 0.5 if that area did not include 0.5 (this figure has been updated per request of Reviewer # 1). We are not sure what the reviewer meant by correlation but the two performance curves are not related to each other and any similarity could be due to similar block length and filtering for the two experiments. Finally, we now report effect sizes (Cohen d values) whenever appropriate.

All these changes are marked with [R2.2] in the revised manuscript.

“(3) Figure 3d and results: The split into trials 1-100 and 100-280 seems to be quite arbitrary and driven by the graphics for calculating a significant difference. Is there any (other) justification for this split and could a more principled split be used?”

Response: This split was not arbitrary and instead was done based on the time course of the performance in Experiment 3. More specifically, we found that in this experiment the performance reached its steady state with the time constant of ~68 trials (see Figure 11

below), and 100 trials was chosen based on this time constant (the performance is about 75% of its steady state at $\sim 1.4 \cdot \tau = 95$ trials).

Nevertheless, to show that our conclusion does not depend on the exact value of the boundary between early and late trials, we computed the difference between the quality of fit of the object-based and feature-based models in early and late trials ($\langle -LL_{object-based} + LL_{feature-based} \rangle_{early} - \langle -LL_{object-based} + LL_{feature-based} \rangle_{late}$) and assigned the boundary at trial 90, 100, and 110 and found the following values (using two-sided sign-rank test):

$\Delta = 0.017 \pm 0.047$, $P = 0.044$, $d = 0.36$ (for boundary at 90)

$\Delta = 0.023 \pm 0.050$, $P = 0.029$, $d = 0.46$ (for boundary at 100)

$\Delta = 0.022 \pm 0.052$, $P = 0.049$, $d = 0.42$ (for boundary at 110)

Moreover, the reported difference was significantly larger than zero (sign-rank test, $P < .05$) for any boundary values between 60 and 130.

We now have mentioned these results in the revised manuscript (see [R2.3]).

Figure 11. Time course of performance in Experiment 3 and its fit using an exponential function (dashed line). The shaded areas indicate s.e.m. and the dashed horizontal line shows the chance level. The time constant for the exponential fit is reported on the plot.

“(4) Several subjects participated in more than one experiment: Was there any influence from previous participation? Did these subjects perform differently (better or worse)? Was within/between subject nature of comparisons accounted for in statistical analyses?”

Response: We had 33 subjects who participated in both Experiments 1 and 2 in a pseudo-randomized order. To test whether there was any effect of previous participation and the order of the experiment (i.e. generalizable first or non-generalizable first, for those who participated in both experiments), we used a general linear model (GLM) to extract the

effect of these two factors on the performance (average harvested reward per trial) of all subjects. The result of this GLM fit did not reveal any effect of previous participation or the order of the experiment (see Table 1 below). Finally, we did not account for repeated subjects in statistical analysis since we did not have any between-experiment comparisons and all our comparisons were within-experiment (e.g. fit of object-based vs. feature-based model, etc.).

These results have been briefly mentioned in the revised manuscript (see [R2.4]).

	Experiment	Order	Previous participation	intercept
Regression coeff.	-0.034±0.007***	0.005±0.008	0.003±0.006	0.599±0.159***

Table 1. Results of the GLM fit did not reveal the effect of previous participation or the order of the experiment on performance. Reported are the values of regression coefficient (mean ± s.e.m.) (***) indicates significance level of $P < .001$). The negative coefficient for Experiment variable (1 for Experiment 1 and 2 for Experiment 2) shows higher performance for Experiment 1 than 2.

(5) Learning rate in experiments 3 and 4: Both learning rates (more inf. and less inf.) seem to be very small for these two experiments (even after exclusion of subjects) and it is unclear whether there really was meaningful learning at all.

Response: We thank the reviewer for pointing this seemingly contradictory result for the estimated learning rates since as shown in Figure 3a and 3e, subjects clearly learned about options during both Experiments 3 and 4 (also reflected in their estimates). There are a few possible explanations for relatively small learning. First, even the learning rates as small as 0.005 would be practical in these experiments since they result in learning on the timescale of $\sim 2/(1/0.005) = 100$ trials (factor 2 is due to coupled learning). More importantly, as mentioned by Reviewer # 1 (see [R1.3] above), what ultimately determines the amount of update is not the learning rate alone but the ratio of the learning rate by the temperature parameter of the sigmoid for choice probability (equivalently learning rate times the weights for each feature) and this relationship becomes more complex since the learning rate and temperature from model fitting are correlated (see Figure 4 above). In other words, a small learning rate for a given feature accompanied by a large weight for that feature can result in significant update of choice behavior due to a single reward outcome.

Importantly, we found that the product of the learning rates and the weights assigned to each feature are larger for the informative compared to the non-informative feature (see Figure 5c-d above). Moreover, the corresponding values of the differential

response, the new measure for capturing the change in behavior due to a single reward feedback (see Figure 5g-h above and [R1.2]), also indicate that most subjects updated their choice significantly due to a single reward feedback. Note that the previous Figure 4 has been revised (see Figure 5 above and new Figure 5 in the revised manuscript) and now shows the results for subjects who adopted feature-based learning only, since learning rates for features for subjects who adopted object-based learning are not very meaningful. Overall, the qualitative match between the analysis based on the product of estimated learning rates and feature weights and the one based on the differential response shows the usefulness of model fitting in revealing important aspects of learning in our experiments.

Some of the aforementioned points have been added to the revised manuscript (see [R2.5]).

“a. Please report effect sizes in addition to p-values.”

Response: Effect sizes are now reported.

“b. Please discuss these low learning rates critically discussed. How can we reliably say something about learning and differentiate between different models of learning if learning is so low?”

Response: This point has been discussed in the response above (see [R2.5]).

“c. Relatedly, is it not a matter of concern that the relative weight on bias is higher than the relative weight on object-based learning throughout the experiment (Fig. 3f)?”

Response: The fact that relative weight of the bias is larger than the relative weight of the object-based approach simply means that the object-based approach is weaker than the constant bias in estimation. At the same time, this means that the feature-based approach is stronger than the constant bias term and thus more prevalent. This is the point we want to make about Experiment 4. However, since this might be confusing to the readers, we have decided to report R^2 (percent variance explained) of the regression model for reward estimates instead of the relative bias over time. This quantity is equal to the percentage of variability in subjects' estimates that can be explained by reward probabilities based on either the feature-based or object-based approaches, or both. The time course of R^2 shows the time course of learning reflected in the subjects' estimates (see Figure 12 below).

Figure 12. Time course of learning and model adoption based on subjects' estimates in Experiments 3 (a) and 4 (b). Plotted are the relative regression weight for object-based approach and the percent variance explained in subjects' estimates (R^2) by the object-based and feature-based reward probabilities.

We have updated the revised manuscript (and corresponding figures) accordingly and have discussed differences between the results for Experiment 3 and 4 more clearly (see [R2.6]).

“(6) Model comparison (Table 1) & Supplementary Figure 1: Are these results directly correlated? In other words, does in every experiment with a low generalizability the estimated error increase and therefore every model not based on the ability to abstract from specific objects (what is the definition here of feature-based) fit better? Please comment on this.”

Response: We thank the reviewer for asking about this relationship. The Supplementary Figure 1 shows that the average error in the estimation of object values based on the average reward values of features decreases as generalizability increases. In other words, “feature-based” approach means approximating object values based on the average values of features using Equation 1. This plot predicts that adopting feature-based models is advantageous in more generalizable environments. The fact that feature-based models provided better fits for more generalizable environments (Supplementary Table 1) presents experimental evidence for this prediction.

A comment about this relationship has been added to the revised manuscript (see [R2.7]).

“(7) Limitations of the new proposed framework are missing in the discussion.”

Response: We thank the reviewer for asking about this. A few limitations of our framework include: a) it does not address the influence of what has been learned on learning strategy and future model adoption; b) it does not address the intermediate steps for transition from feature-based to object-based learning, for example what “conjunctions” of features are constructed and learned over time; and c) it does not address how model adoption depends on the ability of the decision maker to discriminate between feature values (e.g. colors of objects).

We now have discussed these limitations in the revised manuscript (see [R2.8]).

“(8) The model parameters (potentiation and depression rates, level of stochasticity, etc.) appear to be chosen arbitrarily. How plausible are they and can they be corroborated with independent research?”

Response: We thank the reviewer for asking this question. The models parameters were chosen approximately based on the average estimated parameters from subjects’ choice behavior. To address the feasibility of models and chosen parameters (also requested by Reviewer # 1, see [R1.9]), we have simulated all experiments using the same set of parameters (see Figure 8 above). These simulations show that no fine-tuning is needed to qualitatively capture experimental results using our models. We note that qualitatively similar behavior can be obtained using a wide range of model parameters. Finally, we now also show a good match between the observed response and the average simulated differential response using the estimated parameters based on fit of individual subjects’ data (Figure 3 above), indicating that model parameters can be tuned to a new sets of experimental data as well.

Minor points:

“(1) The introduction should be carefully revised. The term of generalizability should be introduced and defined here, since it is a central part of the manuscript (and generalizability index should be defined in the methods section). Also, the description of available frameworks is rather short and doesn’t show the need for a new framework. Additionally, the last paragraph mentions hierarchical decision-making out of the blue. ”

Response: We thank the reviewer for pointing out these issues, which have been

addressed in the revised manuscript (see [R2.9]).

“(2) What are real-world estimates of generalizability and dimensionality?”

Response: This is very good question. We think that in order to deal with almost infinite number of options in the real world, learning the values of options is done separately for different domains (e.g. values of fruits versus ice creams) and thus, feature-based learning occurs semi-independently for different domains. Therefore, the values of dimensionality and generalizability depend on how this “partitioning” is done by a given individual.

This point has been added to the revised manuscript (see [R2.10]).

“(3) What is the learning rate used in Figure 1b?”

Response: We thank the reviewer for pointing out this missing information. We now have added the learning rate value to the figure caption.

“(4) P. 9: “coupled” comes out of the blue”

Response: It is now explained.

“(5) P. 12/Supplementary Fig. 5a: it is unclear in which sense one feature was partially informative. Please provide example/more detail.”

Response: We thank the reviewer for pointing out this missing information, which has been added to the revised manuscript (see [R2.11]).

“(6) Fig. 6 h, I (maybe g): Plots are meant to show HDML-PDML (which would be desirable) but currently seem to show PDML-HDML.”

Response: We thank the reviewer for pointing out this mistake. This has been fixed in the revised manuscript.

“(7) On p. 24, the reference to Kahnt, Chang, Park, et al. 2012 (Journal of Neuroscience) should be replaced with Kahnt, Park, Burke, et al. 2012 (Journal of Neuroscience) or Kahnt & Tobler, 2016 (Elife).”

Response: We thank the reviewer for pointing the correct reference. We fixed the error in reference to Kahnt et al. (2012) and also added Kahnt & Tobler (2016) in the revised manuscript.

“(8) P. 25/26: There may be a perceived discrepancy between the potential need for arbitration stated on p. 25 and the statement that “reward feedback alone can correctly adjust behavior toward a more object-based or amore feature-based approach, without any explicit optimization”. Please clarify”

Response: We thank the reviewer for mentioning these seemingly contradictory statements. The relationship between these statements has been clarified in the revised manuscript (see [R2.12]).

“(9) On p. 26: “The HDML model proposed in this study shares some components with the model of Hunt et al. (2014), though our model includes learning as well. Similarly, Wunderlich et al (2011) also suggested that the brain holds weights for all possible informative dimensions simultaneously, and these weights are updated on every trial.” – Please describe shared components with a bit more specificity.”

Response: This has been clarified in the revised manuscript (see [R2.13]).

“(10) What software has been used for the computer experiments?

(11) What was the age range of the subjects?

(12) Which Matlab version has been used for analysis?”

Response: We thank the reviewer for pointing out these missing pieces of information, which are now added to the Methods.

“(13) Supplementary Figure 2: Superblock length was 196 trials, made up of four blocks of 48 =192 trials. What were the remaining four trials?”

Response: We thank the reviewer for noticing this mistake, which has been fixed in the revised figure.

“(14) Supplementary Figure 3: Given that the maximum performance was considerably higher in the generalizable compared to the non-generalizable experiment, it is unclear why the average performance does not differ/how maximum was defined (in any case, the maximum definition should be described as it used also elsewhere).”

Response: We thank the reviewer for asking for this clarification. First, the performance was different in the two experiments/environments but this difference was masked by the large amount of noise in the previous Supplementary Figure 3. The revised figure (see Figure 6 above) shows the difference in performance between the two experiments more clearly, especially later in the blocks. Second, the maximum performance in the previous Supplementary Figure 3 was computed by assuming that the decision maker selects the more rewarding option on every trial, where the reward values are computed based on the feature-based approach. This was to show how much subjects would lose if they adopted feature-based versus object-based approach in the non-generalizable environment. Note that the maximum performance for adopting object-based approach is similar in the two environments, and equal to that of the feature-based approach in the generalizable environment (Experiment 1). Therefore, in terms of steady state (which what those lines represent), adopting object-based versus feature-based approach is more rewarding in Experiment 2 but not Experiment 1. However, considering reversals, higher adaptability of feature-based learning makes this learning more advantageous in Experiment 1.

The definition of the maximum performance has been added to the captions of all relevant figures to avoid future confusion (see [R2.14]).

(15) Supplementary Figure 3: Please replace “strategies” with “performance” in the last sentence of the figure legend (there could have been different strategies but if so they did not translate into differences in performance).

Response: The wording has been fixed.

“Some typos

(1) P. 5: “how the performance of the two learners depend” -> “depends”

(2) P. 5: “based of features” -> “based on features”

(3) P. 16: add “the” between “any of” and “four experiments”

(4) P. 25: delete “and” between “circuits,” and “arbitration”

(5) P. 49: Legend of Supplementary Figure 3 should refer to Supplementary Figure 2, not

Supplementary Figure 1.”

Response: We thank the reviewer for pointing out these typos, which have been fixed in the revised manuscript.

Reviewer #3:

“The study by Farashahi and colleagues presents behavioural and modelling results addressing the question of how we learn in multidimensional environments – specifically, how we deal with the “curse of dimensionality”. This study addresses a key topic in computational neuroscience and the authors present cleverly designed behavioural experiments combined with elegant modelling – both algorithmic and neural network models to address this issue.”

Response: We thank the reviewer for the positive evaluation of our work.

“Learning the values of individual objects is most precise, but may require many trials when there are several combinations of features - specifically, the number of possible objects is n to the power of m , where m is the number of features (e.g. colour) and n is the number of values for each feature (e.g. blue, red, green). Thus there is a trade-off between accuracy and the time it takes to learn, which becomes a particular issue when environments are non-stable.

Using simulations, the authors first show that an object-based learner outperforms a feature-based learner later during learning, and this is further shifted as a function of generalizability of the environment (i.e. the better actual object reward probabilities can be inferred from feature-based values). Thus, the more generalizable the environment, the more time it takes for an object-based learner to outperform a feature-based learner.

They then present four behavioural experiments. In experiments 1 and 2, the environment was non-stationary. It was found that subjects adopted feature-based learning even though this did not reduce the dimensionality, but only when the environment was generalizable (experiment 1). In contrast, when the environment was non-stationary but did not allow generalisation, subjects adopted an object-based strategy. In experiments 3 and 4, the environment was kept stationary, while increasing the dimensionality of the state space. This allowed them to isolate the influence of dimensionality reduction. Both in experiment 3 and 4, subjects initially adopted feature-based

learning and then shifted to an object-based strategy later on. However, in experiment 4, the increased dimensionality favoured the feature-based strategy – model comparison showed that behaviour was better explained by an object-based learner in experiment 3, whereas the feature-based learner provided a better fit in experiment 4.

Next, the authors set up two neural network models to provide a mechanistic account for the observed learning behaviour, PDML and HDML. In both models, there were two sets of value-coding neurons learn reward values of individual objects (OVE) or features (FVE). Synapses onto both sets of neurons undergo reward-dependent plasticity. The two models differ in terms of how the output of OVE and FVE is used to generate choice. In the PDML, both OVE and FVE feed into a common (final) decision circuit. There is however also a decision circuit that generates a choice based on the output of OVE and FVE, respectively. If a choice results in reward, then the connections between the set that produced the same choice (OVE or FVE) and the final decision circuit are strengthened. By contrast, in the HDML, there is a single selection circuit in which the outputs of OVE and FVE compete to control which element ultimately governs the decision. Performance of the HDML was somewhat better than that of PDML in all four environments. Likewise, the HDML adapted better to differences in volatility or generalizability of the environment.

This is a very elegant study that mechanistically dissects human reinforcement learning. I only have a few points that I would like the authors to clarify.”

Response: Again we thank the reviewer for the positive evaluation of our work, and for providing an excellent summary of our approach and results. We hope that we have clarified the missing points in our response here and in the revised manuscript.

“1) The network modelling results are elegant and show that the HDML and PDML perform the task reasonably well and that the HDML adapts to the key parameters of the environment better than does the PDML. What I am somewhat missing is how the behaviour of the model compares to real subjects. Even if the behaviour is qualitatively similar, it would be good to have a more direct comparison of network model behaviour with experimental data.”

Response: We thank the reviewer for pointing out this shortcoming of our work, which was also mentioned by Reviewer # 1 (see [R1.9]).

To address this issue we simulated choice behavior of the PDML and HDML

models during all four experiments and fit the simulated choice data using the same models used to fit the experiment data (see Figure 8 above). These analyses directly linked our network models' behavior to that of subjects and moreover, revealed the superiority of the HDML model in capturing the pattern of experimental data. More specifically, choice behavior of the HDML model qualitatively matched the pattern of data in all experiments, whereas the PDML model failed to capture choice behavior during Experiment 1.

The above Figure 8 is added as a new Figure 7 to the revised manuscript, and a new paragraph is also added to discuss these results (see [R3.1]).

"2) The algorithmic models were fit to subject data using Nelder-Mead simplex algorithm implemented in Matlab's fminsearch. This appears to be a suboptimal choice to me – this algorithm is notorious for getting stuck in local minima – there are other ways that would certainly be better. Given the low number of parameters in the models, I would suggest at least using a grid search to find an educated starting point for the simplex. Since the models are integral to this manuscript, it is important that the modelling results are immune to any such criticism."

Response: We thank the reviewer for pointing out this problem with fminsearch algorithm in Matlab. To mitigate this problem, we have repeated each fit of experimental data (or simulated choice data) with at least 10 different random initial conditions and picked the best fit among all those fits. Based on the examination of the outcome fits, we found that about 10 initialization is enough to avoid getting stuck in local minima.

This method has been mentioned in the revised manuscript (see [R3.2]).

"3) My understanding is that both in the PDML and HDML, the synapses from both OVE and FVE to the decision circuit undergo reward-dependent plasticity. If this is true, this is not clearly evident from the text and figure 5, where it appears that only the synapses onto OVE/FVE neurons are subject to such plasticity. I would ask the authors to clarify this point."

Response: We thank the reviewer for pointing out this ambiguity in our description of the models. Yes, in the PDML model synapses from OVE and FVE to the final DM circuit undergo reward-dependent plasticity. In the HDML model, synapses from OVE and FVE to the signal-selection circuit are plastic.

We have revised the caption of previous Figure 5 (current Figure 6) and the corresponding text to clarify this ambiguity (see [R3.3]).

“4) The experiments are designed in a very clever and sophisticated manner – which however sometimes makes it difficult for a reader to precisely understand what exactly was done. In particular, I found it hard to grasp from supplementary figure 2 how the different manipulations in terms of dimensionality/generalizability were implemented. In fact, I felt more confused after looking at the figure than before. Can you please think of a way of making this figure and the description easier to follow? Specifically, what is shown in part B – is this different schedules in each line? But then why is there a different symbol to the left on each side? This suggests I should read the plot column-wise – please help this with a clearer description.”

Response: We thank the reviewer for pointing out this ambiguity in the schematic of experimental design. In panel B of Supplementary Figure 2, each column represents a different schedule. We used 8 different schedules (noted with “Rs” etc.) for each of the two environments (experiments). Each row next to a given symbol indicates the reward value associated with that symbol in different schedules.

We now have modified Supplementary Figure 2 and its caption to address this issue (see [R3.4]).

“5) I think there is an important recent paper from Yael Niv’s lab (Leong et al., 2017, Neuron) that also addresses the curse of dimensionality and the propose a two-way interaction between learning and attention. While their contribution is clearly distinct from the approach presented here, I think this study should be given credit here as there are direct links.”

Response: We thank the reviewer for mentioning this recent and relevant paper.

In the revised manuscript, we have provided the link between our study and that of Leong et al. (see [R3.5]).

REVIEWERS' COMMENTS:

Reviewer #1 (Remarks to the Author):

The authors have been very responsive to all the reviewers' suggestions, and I'm glad to see that the new analyses support the results and improve the paper – thank you. I only have two minor suggestions left, regarding my previous point 1a, which lead to new figures S4 and S5.

- figure S4: it would make more sense to plot AIC or BIC than LLH: the point of this analysis is to show that the method used to identify the best model in real subjects is successful at identifying the best model in simulated subjects; and this is done for real subjects with complexity penalization (AIC or BIC), not with LLH.

- Figure S5: this is interesting, but what would make more sense is to test a given model's ability to recover its own parameters (rather than another model's parameters, as plotted here). What is usually done is plotting simulated vs. recovered parameters within each model of interest, and showing that it is close to the diagonal.

Reviewer #2 (Remarks to the Author):

The authors have adequately addressed my previous points. Thank you!

Reviewer #3 (Remarks to the Author):

The authors have performed extensive revisions that include both modifications to the description of the results and methods and additional analyses that, in my view, further strengthen the conclusions drawn by the authors. All in all, I think the manuscript is now in good shape for publication at Nature Communications. Thank you for the thorough revisions and congratulations!

Response to the reviewers' comments, and summary of changes made in response to the comments of the reviewers.

Title: Your favorite color makes learning more adaptable and precise

Authors: Farashahi, Rowe, Aslami, Lee, and Soltani

We are greatly thankful to all the reviewers for their careful reading of our manuscript and for their useful and constructive suggestions. Below, we provide the final point-by-point responses to individual reviewers' comments/concerns.

Reviewer #1 (Remarks to the Author):

"The authors have been very responsive to all the reviewers' suggestions, and I'm glad to see that the new analyses support the results and improve the paper – thank you. I only have two minor suggestions left, regarding my previous point 1a, which lead to new figures S4 and S5."

Response: We thank the reviewer their useful comments that have greatly improved our manuscript.

"- figure S4: it would make more sense to plot AIC or BIC than LLH: the point of this analysis is to show that the method used to identify the best model in real subjects is successful at identifying the best model in simulated subjects; and this is done for real subjects with complexity penalization (AIC or BIC), not with LLH."

Response: We thank the reviewer for pointing out this issue. We now use AIC to compare the goodness-of-fit between different models. As shown in the new Supplementary Figure 4, all our results and conclusions stay the same with the new measure.

"- Figure S5: this is interesting, but what would make more sense is to test a given model's ability to recover its own parameters (rather than another model's parameters, as plotted here). What is usually done is plotting simulated vs. recovered parameters within each model of interest, and showing that it is close to the diagonal."

Response: We thank the reviewer for pointing out this issue. Per reviewer's request, we now plot the estimated versus actual model parameters for the model used to generate the data (see new Supplementary Figure 5).

Reviewer #2 (Remarks to the Author):

“The authors have adequately addressed my previous points. Thank you!”

Response: We thank the reviewer their useful comments that have greatly improved our manuscript.

Reviewer #3 (Remarks to the Author):

“The authors have performed extensive revisions that include both modifications to the description of the results and methods and additional analyses that, in my view, further strengthen the conclusions drawn by the authors. All in all, I think the manuscript is now in good shape for publication at Nature Communications. Thank you for the thorough revisions and congratulations!”

Response: We thank the reviewer their useful comments that have greatly improved our manuscript.